

# On the efficiency of the hybrid and the exact second-order sampling formulations of the EnKF: A reality-inspired 3D test case for estimating biodegradation rates of chlorinated hydrocarbons at the port of Rotterdam

M.E. Gharamti[1,3], J. Valstar[2], G. Janssen[2], A. Marsman[2], and I. Hoteit[1]

[1]King Abdullah University of Science and Technology (KAUST), Thuwal 23955, Saudi Arabia
[2]Deltares, Princetonlaan 6, 3584 CB, Utrecht, The Netherlands
[3]Nansen Environmental and Remote Sensing Center (NERSC), Bergen 5006, Norway

*Correspondence to:* M.E. Gharamti (mohamad.gharamti@nersc.no)

**Abstract.** In this study, we consider the assimilation problem of subsurface contaminants at the port of Rotterdam in the Netherlands. This involves the estimation of solute concentrations and biodegradation rates of four different chlorinated solvents. We focus on assessing the efficiency of an adaptive hybrid ensemble Kalman filter (EnKF-OI) and the exact second-order sampling formulation (EnKF$_{ESOS}$), for mitigating the undersampling of the filter covariance and the observation errors, respectively. A

multi-dimensional and multi-species reactive transport model is coupled to simulate the migration of contaminants within a Pleistocene aquifer layer located around 25 m below mean sea level. The biodegradation chain of chlorinated hydrocarbons starting from Tetrachloroethene and ending with Vinylchloride is modelled under anaerobic environmental conditions for five decades. Yearly concentration data is used, in a synthetic setup, to condition the forecast concentration and degradation rates in presence of model and observational errors. Assimilation results demonstrate the robustness of the hybrid EnKF, for accurately

calibrating the uncertain biodegradation rates. When implemented serially, the adaptive EnKF-OI scheme efficiently adjusts the weights of the involved covariances for each individual measurement. The relevance of the EnKF$_{ESOS}$ is emphasised by a better maintaining of the parameters ensemble spread. On average, a well tuned hybrid EnKF-OI and the EnKF$_{ESOS}$ suggest around 48% and 21% improved concentration estimates and around 70% and 23% improved anaerobic degradation rates, over the standard EnKF respectively.

## 1 Introduction

Subsurface contamination has received significant attention in the last few decades. Consequent cleanup costs have increased the awareness of environmental issues related to contaminated fields (Appelo and Postma, 1994; Drécourt et al., 2006). Historically, it was believed that subsurface contamination could be remediated to natural background contamination levels by digging in the soil and pumping out the contaminated groundwater. However, it was not too long before it was discovered

that there were simply too many contaminated areas to completely remediate. In addition, all available cleaning technologies,



including source removal, are economically not viable to fully resolve the problem (Cunningham and Berti, 1993; Starr and Cherry, 1994; Todd and Mays, 2005).

Governmental authorities are now considering another approach to remediation based on management of industrial groundwater contamination at regional scales. The idea is simply to prevent groundwater contamination from causing negative effects

on humans or ecology, and to control any undesired spreading beyond the boundary of the contaminated site. In the European Water Framework Directive, an option was offered allowing groundwater aquifers to remain contaminated when remediation is too costly and when an adequate monitoring system of the contaminated area is set up (Chave, 2001; Mostert, 2003; Hering et al., 2010). This procedure relies mostly on natural attenuation of contaminant plumes without the need for a direct human intervention. This is often possible given that the size and concentration of dissolved contaminants are frequently subjected to

considerable decline due to natural, and eventually human induced, biodegradation processes. The challenge is then to predict in a cost effective way what type and when contaminants may cause a risk, so monitoring and, if needed, remediation may be undertaken to prevent any unacceptable spreading beyond specific planes of compliance. One efficient way to implement such monitoring system at a regional scale is to use prediction models with monitoring data and combine them using advanced data assimilation techniques (McLaughlin, 2002; Reichle et al., 2002).

Various numerical groundwater contaminant models have been developed in the literature (e.g., Freeze and Cherry, 1979; Pollock, 1994; Dawson et al., 2004; Sun and Wheeler, 2006; Bear and Cheng, 2010). The idea behind forming such models is to simulate and predict the dynamic fluxes and energies, defined as state variables (e.g. groundwater pressure, contaminant concentration), as accurately as possible based on some selected parameters (e.g., porosity, permeability, sorption) that describe the subsurface geometry, fluid and rock properties and surface-subsurface interactions (Moradkhani et al., 2005).

Groundwater contaminant models can be subject to several sources of uncertainties due to poorly known parameters, inputs, and boundary conditions. For instance, we often know very little about the time at which contamination started, the amount of contaminant mass present in a pure phase source zone, the location of the pure phase and the rate at which biodegradation is taking place (Franssen and Kinzelbach, 2009; Gharamti et al., 2013). Other uncertain aspects are the heterogeneity of the parameters such as the hydraulic conductivity, groundwater recharge and the redox state of the groundwater. Therefore, model

predictions of where and when a contaminant crosses a plane of compliance, with what concentration and how long it takes before a pure phase source zone dries up, can be quite uncertain.

One way to reduce uncertainty in model predictions and parameters is to assimilate data into the model. Data assimilation (DA) methods follow a Bayesian formulation by combining prior information of a dynamical system with available measurements to obtain an analysis of the state and parameters (Hoteit et al., 2012; Gharamti et al., 2014a). Sequential DA techniques,

such as the ensemble Kalman filter (EnKF), assimilate the data as it becomes available. The EnKF (Burgers et al., 1998; Evensen, 2003) is a popular DA method in hydrology, operating in consequent forecast and an analysis steps. During the forecast, a set of state realisations is run forward in time using the dynamical model. At the time of the update, a linear Kalman filter (KF) type analysis (Kalman, 1960; Gharamti et al., 2011) is applied to the ensemble members. The EnKF is relatively simple to implement, requiring only forward integrations of the dynamical and observational models. The EnKF has been proven useful

in various subsurface hydrology applications (e.g., Chen and Zhang, 2006; Hendricks Franssen and Kinzelbach, 2008; Zhou



et al., 2011; Li et al., 2012; Crestani et al., 2013; Panzeri et al., 2013; Gharamti and Hoteit, 2014). The parameters most often calibrated are those characterising the flow and the general transport of the contaminants, such as permeability and porosity. Very few applications have tackled the estimation problem of reactive modelling parameters using sequential DA techniques. Bailey et al. (2013) used the EnKF to estimate spatially variable selenium and nitrate reaction rates in near-surface agricultural soil profiles. In another study, Bailey et al. (2012) used the ensemble smoother to infer the denitrification rate constants from synthetic observations of nitrate concentrations.

It is now widely recognised that the performance of the EnKF strongly depends on the ensemble size; a large enough ensemble is required for obtaining good performances. Gharamti et al. (2014b) proposed an efficient hybrid EnKF assimilation scheme for state and parameters estimation, in which the predicted EnKF statistics are complemented with predefined static background covariance in order to mitigate for filter inbreeding and undersampling (Hamill and Snyder, 2000). The hybrid filter was applied to a small-scale synthetic reactive transport model and was found computationally very efficient, providing reliable estimates using fairly small ensembles (50 members). In this study, we test the hybrid EnKF with a realistic large-scale contaminant model and further extend the adaptive hybrid formulation to allow for a serial processing of the observations during the analysis step. For this, the objective functions involved in the adaptive scheme are designed in such a way that the weighting between the background and the filter flow-dependent statistics is adjusted for each assimilated observation.

The stochastic EnKF assimilates perturbed observations during the analysis step in order to (asymptotically) match the second moment of the KF (Burgers et al., 1998). This often introduces noise, which may become significant when the rank of the observational error covariance is larger than the ensemble size (Nerger et al., 2005). Ensemble square root filters, such as the ETKF (Bishop et al., 2001), the SEIK (Pham, 2001; Hoteit et al., 2002), and the DEnKF (Sakov and Oke, 2008) do not require observations perturbations. Yet, the stochastic EnKF tends to "re-Gaussianize" the ensemble distribution, improving the stability of the filter unlike other deterministic schemes that follow the shape of the background distribution (Lawson and Hansen, 2004). In a recent study, Hoteit et al. (2015) proposed a serial EnKF algorithm to mitigate the observation sampling errors in the EnKF. The algorithm, referred to as EnKF with exact second-order observation perturbation sampling (EnKF$_{ESOS}$), is straightforward to implement in any existing serial EnKF code, requiring only removing a single rank from the sample forecast covariance matrix. Compared to the EnKF and the square-root filters, the EnKF$_{ESOS}$ was shown to provide more accurate estimates of the state of a 40-variable Lorenz 96 model. Here, we consider the EnKF$_{ESOS}$ in a realistic large-scale system and further study the impact of mitigating the observation undersampling errors, in the EnKF's analysis, on the state and parameters estimates.

In this paper, we focus on two aspects that are known to limit the efficiency of the EnKF; namely the undersampling of the forecast errors in the forecast step and the observation errors in the analysis step. We consider an industrial groundwater contamination problem at the port of Rotterdam in the Netherlands. Many areas at the port site are contaminated due to various industrial activities. Contamination with chlorinated hydrocarbons (CH) have been detected at the port area. Reductive dechlorination process of four hazardous CH components; namely Tetrachloroethene (PCE), Trichloroethene (TCE), 1,2-Dichloroethene (DCE) and Vinyl Chloride (VC), is believed to be one of the main reactive processes taking place at the port site. We simulate this process using a coupled three-dimensional Flow-Transport-Reaction (3D-FTR) model for a single plume.





The contaminant data collected in 2012 by the municipality of Rotterdam, is used for initialising the contaminant migration, which propagates to surface and deep aquifer layers ($\approx 50$ m below sea level). The goal of the study is to use "synthetic" CH concentration data on a yearly basis, for a total of 50 years, to calibrate four biodegradation rates of the reaction chain. To the best of our knowledge, this is the first study in which biodegradation parameters of a reductive dechlorination process are

estimated in a real-world system using a sequential DA procedure. On top of the biodegradation, the concentrations of the components are also constrained using the EnKF, the hybrid EnKF and the EnKF$_{\text{ESOS}}$ schemes. Sensitivity analyses are performed to study the efficiency and the accuracy of the assimilation schemes under different experimental settings. The filtering schemes are evaluated based on the accuracy of the estimated solute concentrations, the handling of the posterior distributions of the biodegradation rates and computational complexity.

The rest of this paper is organised as follows. Section 2 presents the ensemble assimilation methods. Section 3 describes the large-scale subsurface reactive transport model and its numerical implementation. Section 4 presents the assimilation setup and the experimental scenarios. Results of assimilation experiments are presented and analysed in Section 5. Conclusions and further discussion are given in Section 6.

## 2   The Data Assimilation Framework

The aim of DA is to combine measured observations and a dynamical model in order to compute the best possible estimates of the past, current and future states of the system, together with estimates of the associated uncertainties (Nichols, 2010). We follow the standard discrete nonlinear dynamical system:

$$\mathbf{x}_{k+1} = \mathcal{M}_k\left(\mathbf{x}_k, \boldsymbol{\Theta}_k\right) + \boldsymbol{\eta}_{k+1}, \tag{1}$$

where $\mathbf{x}_k \in \mathbb{R}^{N_{\mathbf{x}}}$ denotes a state vector of $N_{\mathbf{x}}$ variables at time $t_k$, $\boldsymbol{\Theta}_k \in \mathbb{R}^{N_{\Theta}}$ is the vector of model parameters, $\mathcal{M}_k \colon \mathbb{R}^{N_{\mathbf{x}}}$

$\to \mathbb{R}^{N_{\mathbf{x}}}$ is the nonlinear operator that propagates the model state from $t_k$ to $t_{k+1}$. $\boldsymbol{\eta}_{k+1} \in \mathbb{R}^{N_{\mathbf{x}}}$ is a model error (system noise) accounting for model uncertainties. It is commonly assumed that $\boldsymbol{\eta}_{k+1}$ follows a Gaussian distribution $\mathcal{N}\left(\mathbf{0}, \mathbf{Q}_{k+1}\right)$. The measurements obey the following observational system:

$$\mathbf{y}_{k+1} = \mathcal{H}_{k+1}\left(\mathbf{x}_{k+1}\right) + \boldsymbol{\varepsilon}_{k+1}, \tag{2}$$

where $\mathbf{y}_{k+1} \in \mathbb{R}^{N_{\mathbf{y}}}$ is a vector of $N_{\mathbf{y}}$ observations at time $t_{k+1}$, $\mathcal{H}_{k+1} \colon \mathbb{R}^{N_{\mathbf{y}}} \to \mathbb{R}^{N_{\mathbf{x}}}$ is an observational map including grid

interpolations, and could be nonlinear. The observation errors $\boldsymbol{\varepsilon}_{k+1} \in \mathbb{R}^{N_{\mathbf{y}}}$ are Gaussian with zero mean and covariance $\mathbf{R}_{k+1}$. We assume independent model and observation errors.

    Following the Bayesian filtering problem, the objective is to evaluate the joint probability density function (pdf), i.e., $p\left(\mathbf{x}_k, \boldsymbol{\Theta}_k | \mathbf{y}_{0:k}\right)$, of the system state $\mathbf{x}_k$ and the parameters $\boldsymbol{\Theta}_k$ given all available observations $\mathbf{y}_{0:k}$. The observations, $\mathbf{y}_{0:k}$, are used to update the model forecast. The updated estimate is then used to compute a future prediction. Likewise, the estimation

problem can be also tackled using variational approaches that involve minimisation of a cost function (Dimet and Talagrand, 1986; Courtier et al., 1994; Hoteit et al., 2005; Altaf et al., 2013). Variational DA techniques, such as 3DVar and 4DVar, are





widely used in geoscience applications. These methods look for an optimal state trajectory that best fits observational data over a time window, but do not offer an efficient framework for quantifying uncertainties in the solution. In this study, we will only consider the sequential Bayesian filtering problem.

## 2.1 The Ensemble Kalman Filter for State-Parameters Estimation

The computation of $p(\mathbf{x}_k, \boldsymbol{\Theta}_k | \mathbf{y}_{0:k})$ is not feasible in real applications owing to the nonlinear character of the model and observation operators in addition to the very large dimension of the subsurface flow and transport system. The ensemble Kalman filter (EnKF) is an efficient Monte Carlo method, which computes an approximation of the joint pdf at reasonable computational requirements. The EnKF represents the distribution of the system using a collection of joint state and parameters vectors, called ensemble. We follow the state-parameters augmentation procedure (Annan et al., 2005) and denote by $\psi$ the jointly concatenated state and parameters vector. The parameters are time-invariant so that their time-propagation function is simply the identity operator.

To illustrate, starting at time $t_{k-1}$ from an analysis ensemble, $\left\{ \boldsymbol{\psi}_{k-1}^{a,i} : \mathbf{x}_{k-1}^{a,i}, \boldsymbol{\Theta}_{k-1}^{a,i} \right\}_{i=1}^{N_e}$, which represents $p(\boldsymbol{\psi}_{k-1} | \mathbf{y}_{0:k-1})$, the EnKF propagates the dynamical model (1) to compute the forecast ensemble at the time of the next available observation, $t_k$. Incoming measurements are then used to update the joint ensemble. The EnKF algorithm is summarised below.

– Forecast Step: The analysis members are integrated forward in time to obtain the forecast ensemble from which we estimate the first two moments as follows:

$$
\widehat{\boldsymbol{\psi}}_k^f = \left[ \begin{array}{c} \widehat{\mathbf{x}}_k^{f,i} \\ \widehat{\boldsymbol{\Theta}}_k^{f,i} \end{array} \right] = \frac{1}{N_e} \left[ \begin{array}{c} \sum_{i=1}^{N_e} \mathcal{M} \left( \mathbf{x}_{k-1}^{a,i}, \boldsymbol{\Theta}_{k-1}^{a,i} \right) \\ \sum_{i=1}^{N_e} \boldsymbol{\Theta}_{k-1}^{a,i} \end{array} \right] \equiv \frac{1}{N_e} \sum_{i=1}^{N_e} \boldsymbol{\psi}_k^{f,i}, \tag{3}
$$

$$
\widehat{\mathbf{P}}_k^f = \left[ \begin{array}{cc} \widehat{\mathbf{P}}_{xx}^f & \widehat{\mathbf{P}}_{x\theta}^f \\ \widehat{\mathbf{P}}_{\theta x}^f & \widehat{\mathbf{P}}_{\theta\theta}^f \end{array} \right] \equiv \frac{1}{N_e - 1} \sum_{i=1}^{N_e} \left( \boldsymbol{\psi}_k^{f,i} - \widehat{\boldsymbol{\psi}}_k^f \right) \left( \boldsymbol{\psi}_k^{f,i} - \widehat{\boldsymbol{\psi}}_k^f \right)^T. \tag{4}
$$

The joint sample covariance $\widehat{\mathbf{P}}_k^f$ consists, as shown in (4), of the sample state covariance $\widehat{\mathbf{P}}_{xx}^f$, the state-parameters cross-correlation $\widehat{\mathbf{P}}_{\theta x}^f$ and the sample parameters covariance $\widehat{\mathbf{P}}_{\theta\theta}^f$ matrices. The joint state-parameters forecast estimate (mean) is denoted by $\widehat{\boldsymbol{\psi}}_k^f$. The complexity of the forecast step grows with the ensemble size. If one supposes that $\mathcal{C}_M$ is the cost for integrating the model to the next observation time, the computational requirement of the forecast step is $N N_e \mathcal{C}_M$, where $N$ is the final simulation time (Gharamti et al., 2014a).

– Analysis Step: When the observation $\mathbf{y}_k$ becomes available, the joint forecast members $\boldsymbol{\psi}_k^{f,i}$ are updated using the Kalman-update step; i.e.

$$
\boldsymbol{\psi}_k^{a,i} = \boldsymbol{\psi}_k^{f,i} + \mathbf{K} \left( \mathbf{y}_k + \boldsymbol{\epsilon}_k^i - \widetilde{\mathbf{H}}_k \boldsymbol{\psi}_k^{f,i} \right), \tag{5}
$$





where $\mathbf{K} = \widehat{\mathbf{P}}_k^f \widetilde{\mathbf{H}}_k^T \left( \widetilde{\mathbf{H}}_k \widehat{\mathbf{P}}_k^f \widetilde{\mathbf{H}}_k^T + \mathbf{R}_k \right)^{-1}$ is the Kalman gain and the analysis state is:

$$\widehat{\boldsymbol{\psi}}_k^a = \frac{1}{N_e} \sum_{i=1}^{N_e} \boldsymbol{\psi}_k^{a,i} \equiv \widehat{\boldsymbol{\psi}}_k^f + \mathbf{K} \left( \mathbf{y}_k + \widehat{\boldsymbol{\epsilon}}_k - \widetilde{\mathbf{H}}_k \widehat{\boldsymbol{\psi}}_k^f \right), \qquad \widehat{\boldsymbol{\epsilon}}_k = \frac{1}{N_e} \sum_{i=1}^{N_e} \boldsymbol{\epsilon}_k^i. \tag{6}$$

The observation perturbations, denoted by $\boldsymbol{\epsilon}_k$, are sampled from a Gaussian distribution of zero mean and covariance $\mathbf{R}_k$. The observational operator $\widetilde{\mathbf{H}}_k = \left[ \mathbf{H}_k, \mathbf{O} \right]$, acting on the augmented state-parameter vector, is assumed linear for simplicity, and the matrix $\mathbf{O}$ is a zeros matrix. Computationally, the update step in hydrological applications is usually less demanding than the forecast step, with a complexity of $N N_e N_\mathbf{y} N_\mathbf{x} + N N_e^2 \left( N_\mathbf{x} + N_\Theta \right)$. The observations used in the update equation of (5) are processed in one single batch. In our implementation, however, we will consider the serial

EnKF update formulation in which the observations are assimilated one at a time. The reason for this will become clear in section 2.3.

### 2.1.1 EnKF Limitations

The performance of the EnKF strongly depends on the accuracy of the forecast error covariance matrix $\widehat{\mathbf{P}}^f$. The errors in $\widehat{\mathbf{P}}^f$ are essentially due to: (1) model errors and the use of small ensemble sizes, and (2) propagation of errors in the sample

covariance matrix $\widehat{\mathbf{P}}^a$ at the previous step. The Gaussian assumption of the system's distribution is also a limiting factor but this was proven to be less problematic (e.g., Frei and Künsch, 2013).

The main advantage of the ensemble approximation (Eqs. 3 and 4) is that it does not involve any linearisation and allows to represent the first two moments of the state and parameters by an ensemble of vectors (Evensen, 2003). The use of large ensembles is practically not possible and thus the sample covariance $\widehat{\mathbf{P}}_k^f$ may not well approximate the forecast covariance $\mathbf{P}_k^f$

of the KF. As such, the joint forecast pdf of the system's state and parameters at any time $t_k$ is only partially sampled, which means that there exists a null subspace in the error space that is not covered by the ensemble (Song et al., 2010; Mandel et al., 2011). To mitigate this, we will a use hybrid formulation of the forecast state and parameters statistics before performing the EnKF update (e.g., Wang et al., 2007). Further details are given in section 2.2.

The limited ensemble size may also introduce noise in the update step of the EnKF when the rank of the observation

error covariance is large (Hoteit et al., 2015). This is because the number of observation perturbations may not be enough to sample the observation error covariance matrix, $\mathbf{R}_k$. In addition, spurious correlations between the observation and the forecast perturbations will also introduce noise in the EnKF update (e.g., Bowler et al., 2013; Hoteit et al., 2015). To illustrate, the EnKF analysis assumes zero cross-correlations between the observation perturbations and the forecast ensemble; i.e.:

$$\sum_{i=1}^{N_e} \boldsymbol{\epsilon}_k^i \left( \boldsymbol{\psi}_k^{f,i} - \widehat{\boldsymbol{\psi}}_k^f \right)^T = \mathbf{0}. \tag{7}$$

This can be easily seen by subtracting eq. (5) from eq. (6). After arranging the terms and using eq. (4), one obtains:

$$\boldsymbol{\Delta} = \left( \mathbf{I} - \mathbf{K} \widetilde{\mathbf{H}}_k \right) \frac{1}{N_e - 1} \sum_{i=1}^{N_e} \left( \boldsymbol{\psi}_k^{f,i} - \widehat{\boldsymbol{\psi}}_k^f \right) \boldsymbol{\epsilon}_k^{i\,T} \mathbf{K}^T + \mathbf{K} \frac{1}{N_e - 1} \sum_{i=1}^{N_e} \boldsymbol{\epsilon}_k^i \left( \boldsymbol{\psi}_k^{f,i} - \widehat{\boldsymbol{\psi}}_k^f \right)^T \left( \mathbf{I} - \mathbf{K} \widetilde{\mathbf{H}}_k \right)^T, \tag{8}$$

$$\widehat{\mathbf{P}}_k^a = \left( \mathbf{I} - \mathbf{K} \widetilde{\mathbf{H}}_k \right) \widehat{\mathbf{P}}_k^f \left( \mathbf{I} - \mathbf{K} \widetilde{\mathbf{H}}_k \right)^T + \mathbf{K} \mathbf{R}_k \mathbf{K}^T + \boldsymbol{\Delta}, \tag{9}$$





where $\boldsymbol{\Delta}$ is the sampling error term; not accounted for in the EnKF. Consequently, the ensemble analysis covariance matches the optimal KF covariance, $\mathbf{P}_k^a$, only when the observational sampling errors and the cross-correlation terms in $\boldsymbol{\Delta}$ are indeed zero. This can be numerically achieved by assimilating the observations serially using the so-called EnKF with exact second-order perturbations sampling, EnKF$_{\text{ESOS}}$, as will be discussed in more details in section 2.3.

## 2.2 The Hybrid EnKF

The hybrid EnKF and optimal interpolation (EnKF-OI) was introduced as a way to account for small ensemble sizes and model deficiencies in the EnKF (Hamill and Snyder, 2000). Using small ensembles results in rank deficient forecast covariance matrices, that strongly limit the fit to the observations (Song et al., 2010). Neglecting model errors might further produce small ensemble spread, and consequently unrealistic confidence in the forecast (Song et al., 2013). The standard solution for rank deficiency or covariance underestimation is to apply inflation and localization. Inflation artificially inflates the spread of the ensemble around the mean state (Hamill et al., 2001; Hoteit et al., 2002). It is also a simple way to account for neglected model errors (Pham et al., 1998). Covariance localization eliminates spurious correlations by a Schur product multiplication of the under-sampled covariance matrix with a function of local support (Houtekamer and Mitchell, 2001; Sakov and Bertino, 2011). Inflation and localization, although efficient and widely used (especially in atmosphere and ocean application), are generally model dependent and require important tuning efforts. They further do not introduce any new directions to diversify the ensemble, limiting the filter update to a small-dimensional ensemble subspace (Song et al., 2010, 2013). Moreover, global model parameters are not local quantities and therefore localization techniques might not be as straightforward (Devegowda et al., 2007). In addition, the parameters are dynamically constant quantities, and thus large ensembles are usually needed to well approximate the parameters distributions (Hendricks Franssen and Kinzelbach, 2008; Zhou et al., 2012).

The hybrid approach estimates the EnKF's forecast error covariance by a weighted sum of the ensemble covariance and a stationary covariance matrix, typically used in a variational or an optimal interpolation (OI) assimilation system. More specifically, the background state-state and state-parameters covariances are estimated as:

$$\widetilde{\mathbf{P}}_{xx}^{\text{Hybrid}} = \alpha\widehat{\mathbf{P}}_{xx}^{\text{EnKF}} + (1-\alpha)\mathbf{P}_{xx}^b, \tag{10a}$$

$$\widetilde{\mathbf{P}}_{\theta x}^{\text{Hybrid}} = \beta\widehat{\mathbf{P}}_{\theta x}^{\text{EnKF}} + (1-\beta)\mathbf{P}_{\theta x}^b, \tag{10b}$$

where $\widehat{\mathbf{P}}_{xx}^{\text{EnKF}}$ and $\widehat{\mathbf{P}}_{\theta x}^{\text{EnKF}}$ are the sample covariance and cross-correlation matrices of the EnKF ensemble, respectively. The background covariances are denoted by $\mathbf{P}_{xx}^b$ and $\mathbf{P}_{\theta x}^b$, respectively. It was indeed shown by Hamill and Snyder (2000) that this additional stationary background covariance may help representing part of the ensemble's null space that is not represented by the limited ensemble. This procedure is based on physically reliable statistics, although flow-independent, unlike inflation and localization (Wang et al., 2009). The scalar quantities $\alpha$ and $\beta$ are weighting factors, taking values between 0 and 1.

### 2.2.1 Practical Implementation

The static background covariance, $\mathbf{P}_{xx}^b$, is often built on the basis of a long inventory of forecast errors (Wang et al., 2009). It is usually assumed to be of low-rank, $r_x$, and can be factorised into spectral modes using Proper Orthogonal Decomposition





(POD) as follows;

$$\mathbf{P}_{xx}^b = \mathbf{S}\mathbf{\Omega}\mathbf{S}^T = \mathbf{S}\mathbf{\Omega}^{\frac{1}{2}}\left(\mathbf{S}\mathbf{\Omega}^{\frac{1}{2}}\right)^T = \widehat{\mathbf{S}}\widehat{\mathbf{S}}^T, \tag{11}$$

where $\mathbf{S}$ is a matrix of spectral coefficients, $\mathbf{\Omega}$ carries information about the associated spectral variances and $\mathbf{\Omega}^{\frac{1}{2}}$ is the Cholesky decomposition of $\mathbf{\Omega}$. The background perturbation matrix, $\widehat{\mathbf{S}}$, has $r_x$ columns, with $r_x$ often smaller than the number of state variables. The background state and parameters cross-covariance, $\mathbf{P}_{\theta x}^b$, can be also approximated by a low-rank, $r_\theta$,

matrix using singular value decomposition (SVD) if the number of parameters is not equal to the number of state variables (and thus the matrix $\mathbf{P}_{\theta x}^b$ is not square). This decomposition in practice in order to reduce computational burden and memory storage. Accordingly, the complexity of the analysis step (referred to as $\mathcal{O}^a$) in the hybrid EnKF-OI scheme becomes:

$$
\begin{aligned}
\mathcal{O}_{\text{EnKF-OI}}^a &= NN_eN_{\mathbf{y}}N_{\mathbf{x}} + NN_e^2\left(N_{\mathbf{x}} + N_{\mathbf{\Theta}}\right) + NN_e\left(N_{\mathbf{x}}r_x + N_{\mathbf{\Theta}}r_\theta\right), \\
&= \mathcal{O}_{\text{EnKF}}^a + NN_e\left(N_{\mathbf{x}}r_x + N_{\mathbf{\Theta}}r_\theta\right). \tag{12}
\end{aligned}
$$

Given that $r_x$ and $r_\theta$ are usually small in subsurface flow and transport problems (Gharamti et al., 2014b), the complexity of the analysis step of the hybrid EnKF-OI is only marginally larger than that of the EnKF. The complexity of the forecast step of the EnKF and the hybrid EnKF is the same when both are implemented with the same ensemble size.

The weighting factors $\alpha$ and $\beta$ need to be defined in eqs. (10a) and (10b). Careful tuning of $\alpha$ and $\beta$ is very important (Hamill and Snyder, 2000). The simplest way is to select them based on trial and error but this can be computationally very intensive. A

more efficient approach was introduced by Gharamti et al. (2014b) and consists of optimising a one-dimensional (1D) objective function at every update step of the state and the parameters. Based on Kalman's update formulation, assimilating observations causes the uncertainties in the prior estimates to shrink. Thus, using the Kullback-Leibler (KL) divergence (Kullback and Leibler, 1951), one can choose $\alpha$ and $\beta$ that maximise the information gains at the analysis time $t_k$. In this study, we opt to assimilate the observations serially and thus one can adaptively compute optimal weighting factors as follows:

$$
\begin{aligned}
\quad \underset{\alpha}{\arg\max}\ \mathcal{F}(\alpha) &= \underset{\alpha}{\arg\max}\ tr\left[\widetilde{\mathbf{P}}_{xx}^f - \widetilde{\mathbf{P}}_{xx}^a\right], \\
&= \underset{\alpha}{\arg\max}\ tr\left[\widetilde{\mathbf{P}}_{xx}^f\mathbf{H}^T\left(\mathbf{H}_k\widetilde{\mathbf{P}}_{xx}^f\mathbf{H}_k^T + \mathbf{R}_k\right)^{-1}\mathbf{H}_k\widetilde{\mathbf{P}}_{xx}^f\right], \\
&\overset{\text{single observation}}{\equiv} \underset{\alpha}{\arg\max}\ \frac{1}{d}\sum_{m=1}^{N_{\mathbf{x}}}\left(\mathbf{c}_{xx}^{[m]}\right)^2, \tag{13}
\end{aligned}
$$

where $tr\left[\cdot\right]$ denotes the trace of a matrix and $d$ is a scalar quantity equivalent to observation variance $\left(\mathbf{H}_k\widetilde{\mathbf{P}}_{xx}^f\mathbf{H}_k^T + \mathbf{R}_k\right)$ when assimilating one observation. $\mathbf{c}_{xx}^{[m]}$ is the $m^{\text{th}}$ forecast variance-component corresponding to one observed variable. Similarly,

one can derive the objective function for the parameters' weighting factor as follows:

$$
\begin{aligned}
\underset{\beta}{\arg\max}\ \mathcal{G}(\beta) &= \underset{\beta}{\arg\max}\ tr\left[\widetilde{\mathbf{P}}_{\theta\theta}^f - \widetilde{\mathbf{P}}_{\theta\theta}^a\right], \\
&\overset{\text{single observation}}{\equiv} \underset{\beta}{\arg\max}\ \frac{1}{d}\sum_{m=1}^{N_{\mathbf{\Theta}}}\left(\mathbf{c}_{\theta x}^{[m]}\right)^2, \tag{14}
\end{aligned}
$$





where $\mathbf{c}_{\theta x}^{[m]}$ is the forecast cross-correlation component between the $m^{\text{th}}$ parameter and one observed variable. Such KL criterion describes the information gain from each individual observation as it reflects the difference between the prior and the posterior distributions. The interesting point here is that for each observation, different weights would be assigned to the background and the ensemble statistics. The maximisation problems in (13) and (14) are 1D and bounded, yielding minimal forecast variance after the update. In terms of implementation, we perform the optimisation, over the interval $[0,1]$, using a computationally efficient scheme combining both golden-section search and repeated parabolic interpolation (Forsythe et al., 1977).

### 2.3 Exact Second-Order Observation Perturbations Sampling

The sampling error from neglecting the cross-correlation terms in eq. (9) in the EnKF analysis is generally not globally small. It is often composed of a large number of elements that can add up after successive assimilation steps (Hoteit et al., 2015). This may degrade the filter's accuracy and increases the underestimation of the analysis error covariance (Whitaker and Hamill, 2002). Furthermore, such sampling errors can propagate to subsequent steps, eventually deteriorating the performance of the filter.

In a mathematical sense, for the condition in eq. (7) to hold, the rank of the forecast perturbation matrix

$$\Psi_k^f = \left[ \boldsymbol{\psi}_k^{f,1} - \widehat{\boldsymbol{\psi}}_k^f, \boldsymbol{\psi}_k^{f,2} - \widehat{\boldsymbol{\psi}}_k^f, \ldots, \boldsymbol{\psi}_k^{f,N_e} - \widehat{\boldsymbol{\psi}}_k^f \right] \tag{15}$$

plus the rank of $\mathbf{R}_k$ must not exceed $N_e - 1$, which is essentially the rank of $\Psi_k^f$ (Pham, 2001). Obviously, this is not possible given that $N_{\mathbf{y}} + N_e - 1$ is always greater than $N_e - 1$. Yet, if we suppose that $\Psi_k^f$ has a rank $N_e - 2$, then when $\mathbf{R}_k$ is scalar, it is possible to draw the observation perturbations $\epsilon_k^i$ such that the EnKF's analysis first and second moments are exactly the same as those computed using the KF. Accordingly, Hoteit et al. (2015) proposed to remove one rank from $\Psi_k^f$ using an SVD decomposition:

$$\boldsymbol{\psi}_k^{f,i} \leftarrow \boldsymbol{\psi}_k^{f,i} - \left( \Psi_k^f \mathbf{w}_k \right) w_k^i, \tag{16}$$

where $\mathbf{w}_k$ is the normalised right singular vector of $\Psi_k^f$ associated with the smallest nonzero singular value. The $i^{\text{th}}$ component of $\mathbf{w}_k$ is denoted by $w_k^i$ and the symbol $\leftarrow$ means "replaced by." Then, assimilating the observations serially and simply choosing $\epsilon_k^i = \sqrt{(N_e - 1)\mathbf{R}_k} w_k^i$ would guarantee zero cross-correlations between the modified forecast perturbations and the observation perturbations. The algorithm, referred to as EnKF$_{\text{ESOS}}$, involves a recursive update for $\mathbf{w}_k$ after each update. The serial analysis procedure of the EnKF$_{\text{ESOS}}$ is summarised in the algorithm below:

**While** $\quad j$ in $1,\ldots,N_{\mathbf{y}} \quad$ **do**

1: $\widehat{z}_k = \mathbf{H}_k^{[j]} \widehat{\boldsymbol{\psi}}_k$

2: $z_k^{[j],i} = \mathbf{H}_k^{[j]} \widehat{\boldsymbol{\psi}}_k^{a,i}$

3: $\mathbf{K}^{[j]} = \sum_{i=1}^{N_e} \left( \boldsymbol{\psi}_k^{f,i} - \widehat{\boldsymbol{\psi}}_k^f \right) \left( z_k^{[j],i} - \widehat{z}_k \right) \left[ \sum_{i=1}^{N_e} \left( z_k^{[j],i} - \widehat{z}_k \right) + (N_e - 1)\mathbf{R}_k^{[j,j]} \right]^{-1}$

4: **For** $\quad i$ in $1,\ldots,N_e \quad$ **do**





$$\psi_k^{a,i} \leftarrow \psi_k^{a,i} + \mathbf{K}^{[j]}\left(\mathbf{y}_k^{[j]} + s\sqrt{(N_e-1)\mathbf{R}_k^{[j,j]}}w_k^i - z_k^{[j],i}\right)$$

5: **EndFor**


6: $\displaystyle \widehat{\psi}_k^a \leftarrow \frac{1}{N_e}\sum_{i=1}^{N_e}\psi_k^{a,i}$

7: **For** $\quad i$ in $1,\ldots,N_e \quad$ **do**

$$w_k^i \leftarrow s\sqrt{(N_e-1)\mathbf{R}_k^{[j,j]}}w_k^i - \left(z_k^{[j],i} - \widehat{z}_k\right)\left[\sum_{i=1}^{N_e}\left(z_k^{[j],i} - \widehat{z}_k\right)^2 + (N_e-1)\mathbf{R}_k^{[j,j]}\right]^{-\frac{1}{2}}$$

8: **EndFor**

**EndWhile**

where $s$ is an independent plus or minus sign. The superscript $[j]$ denotes the $j^{\text{th}}$ element and row of the given vector and matrix, respectively. The superscript $[j,j]$ denotes the element in row and column $j$ of the associated matrix. Note that unlike the EnKF, the observation perturbations cannot be Gaussian because of the constraint they satisfy in eq. (7). In the experiments of Hoteit et al. (2015), these were shown to be almost Gaussian. In term of complexity, the EnKF$_{\text{ESOS}}$ algorithm has almost the same computational cost as that of the serial EnKF. Additional cost is required for iteratively updating the vector $\mathbf{w}_k$ and

performing an SVD on $\boldsymbol{\Psi}_k$ to reduce its rank by one. Both operations are computationally almost negligible compared to the cost of integrating the subsurface model.

## 3   The Subsurface Model and Assimilation Experiments

### 3.1   The Rotterdam Port and Geology of the Area

The Port of Rotterdam is located in the Netherlands between the city of Rotterdam and the North Sea. It is the largest port

of Europe covering an area of 105 km$^2$ and stretching over a distance of 40 km. The original geology of the area consists of a top Holocene layer of approximately 20 m thick (Figure 1). It is composed of clay and peat with local sandy channel deposits, but in the most western part, it becomes sandier. Under the Holocene layer, there is a Pleistocene aquifer of coarse sand of approximately 10 m thick. Below lays a Pleistocene clay layer of approximately 30 m thick and a second aquifer of approximately 140 m thick. The second aquifer is saline for most of the port areas whereas the first aquifer is partly saline in

the western part only. On top of the Holocene sediments, an anthropogenic layer of fine sand was added up to a level of 4 m (eastern part) to 6 m (western part) above the mean sea level. Moreover, locally a dense network of sand filled vertical drains was used in the upper part of the Holocene clay in order to speed up the settling of the clay. A large part of the industrial port area is surrounded by surface water, some of which continue to the bottom of the Holocene layer.

At the port site, more than 600 companies perform various activities such as trans-shipment of containers (coal, oil, gas, etc),

storage of oils and chemicals, building/repairing ships and oil/gas rigs, distribution and transport inland, and disposal/treatment of chemical wastes. As a result of the long-term presence of these industrial activities, the soil and groundwater have become





contaminated. This contamination is substantial, complex, and not limited to one particular site but affects the groundwater systems at a regional scale (Marsman et al., 2006; Ter Meer et al., 2007). Part of the contaminants are non-mobile such as heavy metals including arsene, cadmium, copper, mercury, lead and zinc. Other mobile contaminants are mineral oils, volatile

aromatics, chlorinated solvents and pesticides.

## 3.2 Coupled 3D Subsurface Model

### 3.2.1 Organic Contaminants

Wells monitoring and lab analysis have concluded that groundwater at the port area is contaminated, at different depth, with varying levels of pollutants (Marsman et al., 2006). One of the major contaminants are chlorinated hydrocarbons that

had entered the subsurface as Dense Non-aqueous Phase Liquids (DNAPL) and often have source zones of stagnant pure phases at considerable depth. Numerous industrial companies at the port manufacture or work with these organic molecules. Here, we simulate the degradation chain of four CH components; namely Tetrachloroethene (PCE, a.k.a perchloroethene), Trichloroethene (TCE), 1,2-Dichloroethene (DCE) and Vinyl Chloride (VC). We use plume data from a real site, but for confidentiality reasons we do not show the exact location of the site. The horizontal area of the domain is equal to 1.5 km$^2$, extending

1 km in the transverse direction and 1.5 km in the longitudinal direction (Figure 2). Degradation of the dissolved components takes place as chlorine atoms are subsequently replaced by hydrogen atoms under anaerobic environmental conditions (Vogel and McCARTY, 1985; Clement et al., 2000; Tobiszewski and Namieśnik, 2012). Chlorinated hydrocarbons can pose serious threat to human and environmental health (Ojajärvi et al., 2001; Lee et al., 2002, 2003).

### 3.2.2 Flow-Transport-Reaction Model (FTR-Model)

The subsurface model consists of three major components; namely flow, transport and reactions. First, the groundwater flow (assumed steady) is solved on a rectangular domain using MODFLOW (Harbaugh, 2005). Next, MT3DMS is used to solve the advection-dispersion based transport of the components (Zheng and Wang, 1999), in which the degradation process of the components is added based on the module within the 3D-multispecies reactive package; RT3D (Clement, 1997). The softwares are integrated in a sophisticated fortran-based tool (with graphical interface) called iMOD (Vermeulen et al., 2013).

In differential form, the fate and transport of the components is modelled following:

$$\left(\phi + \rho_b k^\ell\right) \frac{\partial \mathbf{C}^\ell}{\partial t} + \lambda \phi \mathbf{C}^\ell = \nabla \cdot \left(\phi \mathcal{D} \nabla \mathbf{C}^\ell\right) - \nabla \cdot \left(\nu \mathbf{C}^\ell\right) + q_s \mathbf{C}_s^\ell + r_{\mathbf{C}}, \tag{17}$$

where $\phi$ is porosity, $\rho_b$ is the bulk density of the soil, $k$ is the distribution (sorption) coefficient, $\mathbf{C}$ is the solute concentration, $\lambda$ is first-order reaction rate, $\mathcal{D}$ consists of hydrodynamic dispersion and molecular diffusion, $\nu$ denotes the Darcy velocity, $q_s$ is the volumetric source/sink flow rate, $\mathbf{C}_s$ is the source/sink flux concentration and $r_{\mathbf{C}}$ refers to the rate of reactions.

The superscript $\ell$ corresponds to the component number taking values between $1$ and $4$ in this study. Along with the basic groundwater flow and transport equations, and using the reaction operator-split strategy (Clement et al., 1998), the biological





reaction kinetics are assembled as a set of ordinary differential equations as follows:

$$\frac{\partial \mathbf{C}_{\text{PCE}}}{\partial t} = -\frac{K_P \mathbf{C}_{\text{PCE}}}{R_P}, \tag{18a}$$

$$\frac{\partial \mathbf{C}_{\text{TCE}}}{\partial t} = -\frac{1}{R_T}\left(K_T \cdot \mathbf{C}_{\text{TCE}} - S_{T/P} \cdot K_P \cdot \mathbf{C}_{\text{PCE}}\right), \tag{18b}$$

$$\frac{\partial \mathbf{C}_{\text{DCE}}}{\partial t} = -\frac{1}{R_D}\left(K_D \cdot \mathbf{C}_{\text{DCE}} - S_{D/T} \cdot K_T \cdot \mathbf{C}_{\text{TCE}}\right), \tag{18c}$$

$$\frac{\partial \mathbf{C}_{\text{VC}}}{\partial t} = -\frac{1}{R_V}\left(K_V \cdot \mathbf{C}_{\text{VC}} - S_{V/D} \cdot K_D \cdot \mathbf{C}_{\text{DCE}}\right), \tag{18d}$$

where $\mathbf{C}_{\text{PCE}}, \mathbf{C}_{\text{TCE}}, \mathbf{C}_{\text{DCE}}$, and $\mathbf{C}_{\text{VC}}$ are the concentrations of the components, $K_P, K_T, K_D$, and $K_V$ are first-order anaerobic degradation rate constants, $S_{T/P}, S_{D/T}$, and $S_{V/D}$ are stoichiometric yield values, and $R_P, R_T, R_D$, and $R_V$ are retardation factors. Linear sorption conditions are assumed for all components.

The model domain as indicated by the blue region of Figure 2 is discretised horizontally into $20 \times 30$ grid cells of $50 \times 50$ m. In the vertical direction, we consider 120 layers each of 0.5 m thickness. The discretisation is based on the geological voxel model GeoTOP (Stafleu et al., 2011). The top layer starts at 7.5 m above sea level, whereas the lowest layer is located at around 52.5 m below sea level. Based on different simulations conducted as part of this study, the migration of the contaminants was found to be limited to a certain depth. We thus assume that only layers $21 - 100$ are active. Figure 2 also shows the contaminant source (in yellow) consisting of four CH components with uniform concentration values. The plume data was obtained in January 2012 from a depth of 22.5 m below mean sea level (model layer 60), in which $C_{\text{PCE}} = 1083.0, C_{\text{TCE}} = 238.0, C_{\text{DCE}} = 633.0$, and $C_{\text{VC}} = 833.0$ µg/l. This contaminant plume is considered as the initial condition of the transport simulations in this study. Furthermore, the PCE plume is used as a continuous contamination source and was included in the Source/Sink Mixing [SSM] package of the MT3DMS simulator. Up to this date, other time-series and well contaminant data are not accessible due to confidentiality imposed by local companies. Modelling parameters required for running the coupled FTR-Model, such as porosity, distribution coefficients and others are defined, based on real data and laboratory assessment, as 3D heterogeneous fields. In Table 1, we report the mean value (averaged over all layers) for some of these parameters. As an illustration, we show in Figure 3 the spatial map of the distribution coefficient of TCE averaged over the top 10 layers. The map shows larger sorption degrees in the northeast part of the domain. This gradually decreases towards the southern region.

## 3.3 Assimilation Experiments

### 3.3.1 Reference Run and Pseudo-Observations

In the scope of twin-experiments, we first conduct a reference model run using some "true" (reference) parameters and initial condition. Next, we impose different uncertainties on the model and the initial conditions, and we assimilate pseudo-observations extracted from the reference run to recover the "true" trajectory of the model. The goal is to estimate the concen-





tration of chlorinated hydrocarbons (i.e., state variables) and their associated degradation rates (i.e., parameters):

$$\mathbf{x} = \left[ \begin{array}{cccc} \mathbf{C}_{\text{PCE}}^T, & \mathbf{C}_{\text{TCE}}^T, & \mathbf{C}_{\text{DCE}}^T, & \mathbf{C}_{\text{VC}}^T \end{array} \right]^T, \tag{19a}$$

$$\boldsymbol{\Theta} = \left[ \begin{array}{cccc} K_P, & K_T, & K_D, & K_V \end{array} \right]^T. \tag{19b}$$

Based on the model's configuration, the dimension of the state is $N_{\mathbf{x}} = 288,000$ and the parameters $N_{\boldsymbol{\Theta}} = 4$. The reference values of the anaerobic degradation rates are obtained through field and laboratory testing (Suarez and Rifai, 1999) and are

given as $K_P = 0.068$, $K_T = 0.086$, $K_D = 0.004$, and $K_V = 0.153$ per day. We perform the reference run for a 50-years period using these rates along with the parameters of Table 1 and the initial conditions, $\mathbf{x}_0$, as defined above. We plot the resulting hydraulic head field at four different layers (i.e., 30, 50, 70, and 90) in Figure 4. The maps clearly show the southward and downward flow direction of the groundwater. The hydraulic head varies between 1.5 m in the center of the domain and drops to around -1 m in the southern part. The top layers, on average, have larger hydraulic heads than the deeper ones. Overall, the

flow configuration indicates that the contaminant plume would follow the behaviour of the groundwater and predominantly moves vertically downwards and laterally in the southwards direction.

Following this steady flow, we then simulated the reference transport of PCE, TCE, DCE, and VC. The time step of the transport-reaction simulator was about 11 days. To visualise the migration process, we show in Figure 5 the contaminant evolution of PCE, TCE, DCE and VC in layers 40, 60, 80, and 100 after 50 years, respectively. As shown, the contaminant plume,

which is originally present in layer 60, has moved into deeper Pleistocene layers. After 50 years, the maximum concentration of DCE in layer 80 reaches 650 $\mu$g/l. Careful assessment of the transport process shows that the four plumes have reached the last active layer in the second aquifer; i.e., layer 100. This is mostly due to the continuous PCE contamination source located in layer 60. Contaminant concentrations in the top Holocene layers are much smaller. Laterally, the contaminant plume is seen to expand from its initial location to a distance of 1.3 km southwards.

From the reference run, we collect pseudo-observations of the concentration to use them later for assimilation. Observations are assumed available for all components from layers 30, 50, 70, and 90. From each of these four layers, only 10 data points are collected and thus a total of $N_{\mathbf{y}} = 160$. Note that $N_{\mathbf{y}}$ is much smaller than the number of state variables, $N_{\mathbf{x}}$. This is usually the case in subsurface hydrology applications, given the significant and expensive cost incurred for preparing, drilling, and completing wells. The observation points are uniformly distributed throughout the domain as denoted by green triangles

in Figure 2. We assume that these 160 measurements are available for assimilation on a yearly basis. We also place a control well in layer 70 around the center of the domain, particularly at the local coordinates $x = 450$ and $y = 600$ m, to monitor the concentration evolution in time. We further assume that these observations are noisy, in order to mimic realistic settings. We thus perturb them with a Gaussian noise of mean 0 and standard deviation equal to 15% of the total observation mean (averaged over the entire 50 years). During assimilation, the updated concentration values are monitored to make sure that they

are non-negative. Cell values that fall out of this constraint are set to zero to obtain a physically meaningful solution (Li et al., 2012; Gharamti et al., 2013).



### 3.4 Initial Ensemble and Background Statistics

To initialise the filters, we perform an unconditional 50-years simulation run (referred to as free run) starting from the mean concentration of the reference model run. Thus, at time $t = 0$ the concentration of the four CH components is not only present in layer 60, but rather spread-out in all layers. In this free model run, we use around $30\%$ larger degradation rates than the reference values. The concentration of the components was saved each 6 months. Next, we randomly select a set of $N_e$ concentration snapshots from the free run outputs to form the state ensemble. This prior (initial) ensemble is quite far from the truth and further has a relatively small spread. This is chosen in purpose to test the robustness of the assimilation schemes to challenging initial uncertainties. The initial parameters ensemble is sampled assuming a Gaussian distribution with mean equal to the reference rate values and variance 40%.

The background error statistics required for the EnKF-OI scheme are parameterised as follows. We form a set of 200 degradation realisations, as described above, and use these to perform 3-months forecasts starting from a series of initial concentrations distributed at 3-months intervals over a 50-years period, as outlined in Figure 6. To illustrate, starting from the mean concentration of the reference run, one realisation of the degradation rates $\boldsymbol{\Theta}_0$ is used to obtain a 3-months forecast of the concentration $\mathbf{x}_1$. Then, using $\mathbf{x}_1$ and $\boldsymbol{\Theta}_1$, the 3D-FTR model is integrated forward to obtain $\mathbf{x}_2$ concentration after 6 months. We continue this process until the end of the 50 years period. Then, we collect the predicted contaminant states for all components and augment them with the corresponding degradation rates in a joint matrix form. POD and SVD are then performed on the augmented concentration-degradation forecast perturbations to summarise the correlations by a small number of orthogonal patterns (Hoteit et al., 2002; Skachko et al., 2009; Altaf et al., 2013). Consequently, the parameterisation of the background covariance matrix, $\mathbf{P}_{xx}^b$, is achieved using the leading 10 POD modes (i.e., $r_x = 10$) of this ensemble, which capture more than $98\%$ of the total variance. Concerning the background cross-correlations, we use the first 10 singular vectors (thus, $r_\theta = 10$) to parametrise $\mathbf{P}_{\theta x}^b$ matrix. To visualise these correlations, we plot in Figure 7, as an illustration, the spatial correlation between the rate at which PCE is degrading and the concentration of the ending product of the chain, VC. Clearly, VC concentration in layer 60 exhibits the largest correlation values because of the continuous source term. Furthermore, since the groundwater flow is stronger in the downwards direction, and so is the contaminant migration, the cross-correlations in deeper layers are larger than those of the shallow layers. The background $\mathbf{P}_{\theta x}^b$ seem to vanish in the upper parts of the Holocene clay and peat layer. A consistent behaviour is observed for the remaining three degradation rates, in which the largest correlations are those associated with $C_{\text{PCE}}$ and smallest with $C_{\text{TCE}}$. PCE has the highest correlation because of the continuous source zone of PCE. Any removal of PCE due to biodegradation in the source zone is directly replenished the next time step, and therefore $K_P$ determines the total load of chlorinated hydrocarbons in the system. On the other hand, TCE has the lowest correlation because its value is high and that makes the parameter relatively insensitive to the amount of biodegradation taking place as compared to the other degradation rate constants.





State and parameters estimates of the EnKF, hybrid EnKF-OI and EnKF$_{\text{ESOS}}$ schemes are evaluated using two metrics; namely mean-squared-error (MSE) and average-ensemble-spread (AES):

$$\text{MSE} = N_{\mathbf{x}}^{-1} N_e^{-1} \sum_{j=1}^{N_e} \sum_{i=1}^{N_{\mathbf{x}}} \left( \mathbf{x}_{j,i}^e - \mathbf{x}_i^t \right)^2, \tag{20a}$$

$$\text{AES} = N_{\mathbf{x}}^{-1} N_e^{-1} \sum_{j=1}^{N_e} \sum_{i=1}^{N_{\mathbf{x}}} \left| \mathbf{x}_{j,i}^e - \hat{\mathbf{x}}_i^e \right|, \tag{20b}$$

where $\mathbf{x}_i^t$ is the true value of the variable at location $i$, $\mathbf{x}_{j,i}^e$ is the forecast ensemble value and $\hat{\mathbf{x}}_i^e$ is the corresponding ensemble mean. MSE measures the distance from the estimate to the truth and AES measures the spread or the uncertainty of the estimates (Hendricks Franssen and Kinzelbach, 2008).

## 4  Results and Discussion

In this section, we present and compare assimilation results with the Rotterdam port's 3D-FTR model using the standard EnKF, the hybrid EnKF-OI and the EnKF$_{\text{ESOS}}$ schemes. The observations are assimilated serially in all three schemes. Concentration and degradation rate estimates of the filters are compared in terms of accuracy and spread. Concentration data are assimilated every year for a total of $50$ assimilation cycles. The ensemble size, $N_e$, is set to $48$ in which batches of four members are run in parallel using Fortran's OpenMP library.

### 4.1  The Hybrid EnKF-OI vs the EnKF

#### 4.1.1  Adjusting Concentration Statistics

To initiate the assimilation experiments, we first run the EnKF and the hybrid EnKF-OI, implemented using only state background statistics, i.e., using eq. (10a) and $\beta = 1$. We carry out 10 different experiments by changing the weighting factor $\alpha$, for each individual run, between 0 to 1 with a step of $0.1$. To visualise the resulting estimates, we plot the average MSE of the contaminant concentrations, averaged over the 4 components and in time, in Figure 8. As shown in the left panel of Figure 8, the most accurate concentration estimates are obtained using $\alpha = 0.7$. This indicates that out of the total forecast error variance, the best reconstruction of the reference contaminant solution is obtained when $30\%$ of this variance is traced from the background statistics. Increasing or decreasing this background contribution (i.e., $30\%$) results in less accurate contaminant estimates. We also note that the least accurate estimates are those obtained when $90\%$ of the ensemble statistics are built based on the static background error covariance. On average, we found that when $\alpha$ takes values between $0.4$ and $0.9$, the EnKF-OI is 16% more accurate than the EnKF.

We also study the effect of changing $\alpha$ on the resulting parameters' estimates. In the right panel of Figure 8, we plot the average MSE of each individual degradation rate obtained using the EnKF and 9 different EnKF-OIs (i.e., $\alpha = 0.1, 0.2, \ldots, 0.9$). We notice that the most influenced biodegradation rates are those associated with TCE and VC. In fact, $K_T$ and $K_V$ are the least identifiable parameters and therefore a small difference in the estimation algorithm (i.e., hybrid EnKF-OI and the EnKF)



may lead to different estimates. In contrast, $K_P$ and $K_D$ are less sensitive to the weighting factor $\alpha$. In accordance with the estimates of the contaminant concentrations, the best match for $K_T$ is obtained using $\alpha = 0.7$. This is not the case for the other parameters; in which $\alpha = 0.3, 1$ and $0.8$ resulted in the best fit to the reference degradation rates of VC, PCE and DCE,

respectively. On average, $K_T$ and $K_V$ estimates are 13% and 35% more accurate than those of the EnKF, respectively. The key point is that complementing the state statistics, using a weighted error covariance as in (10a), does not only contribute to a better retrieval of the concentration but also helps adjusting the cross-correlations with the uncertain parameters. This is essentially the case when biodegradation is taking place at a higher rate, as in $K_T$ and $K_V$, and thus the more information fed through observations, the better the state-parameters cross-correlations would be.

To better understand the performance of the EnKF-OI scheme, we further study how the uncertainties of these degradation rates are maintained in time. For this, we plot in Figure 9 the overall ensemble spread of the four degradation rates obtained using the EnKF and the EnKF-OI (all tested $\alpha$'s). As shown, the EnKF's spread around these 4 parameters is quickly reduced after the first 2 or 3 years. This rapid reduction of the ensemble spread, which is due to the relatively small ensemble size and large initial uncertainties, limits the ability of the filter to impose larger corrections in the future, eventually degrading the

accuracy of the estimated parameters. In contrast, the EnKF-OI maintains larger uncertainties in time for different weighting factor values. As such, as $\alpha$ decreases from 1 to 0.1 the performance approaches that of the EnKF. For instance, after the second year, the ensemble spread of the EnKF reaches 0.03 while it continues to be higher for the EnKF-OI and equal to 0.09, 0.07 and 0.045, respectively. This in turn helps the hybrid filter benefit more from the assimilation of concentration information. This can be confirmed by noticing that the spread of the hybrid filter continues to decrease after 25 years of assimilation, unlike

the EnKF that does not show signs of more corrections. All in all, selecting $\alpha = 0.7$ seems to maintain enough spread for both components' concentration and degradation rates and it retains, on average, the most accurate estimates. For the sake of comparison, we refer hereafter to this scenario as EnKF-OI$_{\alpha=0.7}$.

### 4.1.2 Adjusting Concentration and Parameters Statistics

In the following set of experiments, we fix $\alpha$ to 0.7 and focus on changing the weighting factor between the EnKF and

background state and parameters cross-correlations; i.e., $\beta$. As in the previous section, we conduct 9 experiments in which we change $\beta$ between 0.1 and 0.9. Note that the larger $\beta$ is, the closer the performance is to EnKF-OI$_{\alpha=0.7}$. To analyse the results, we plot in Figure 10 the average MSE and AES of the chlorinated hydrocarbon concentrations. Compared to the previous runs that hybridise the state only, including background cross-correlations information slightly increases the spread around the ensemble mean of concentration, as observed for $0.1 < \beta < 0.3$. In terms of accuracy, varying $\beta$ between 0.1 and 0.6

yields more accurate concentration estimates for all components. To illustrate, when $\beta = 0.1$ the average improvements over the EnKF and the EnKF-OI$_{\alpha=0.7}$ are around 50% and 32%, respectively. This vigorous performance suggests that using only 10% of the "flow-dependent" parameters' ensemble to characterise the pdf of the system is enough to outperform the EnKF. In essence, the background state and parameters cross-correlations seem to carry sufficient description of how the degradation rates and the concentration of each of the components are related. Consequently, only a small portion (i.e., 10%) of the online

parameters' ensemble is required to obtain an accurate biodegradation picture, while the rest of the information all comes from





the specified offline background statistics. This could be due to the time-independent nature of the propagation step describing the evolution of the degradation rates, thereby manifesting a minimal dependence on the online ensemble. This observation comes in accordance with the famous steady-state Kalman filter (El Serafy and Mynett, 2008) that assumes time-invariant error covariance matrix as long as accurate spatial correlations are used within the so-called Kalman gain. In here, our experimental

results suggest that the best parameter's hybrid covariance matrix is very close to a steady-state one. However, this is only for the parameters and this was not the case for the state as described in section (4.1.1). Following the notation introduced earlier, we refer to this scenario, hereafter, as EnKF-OI$_{\alpha=0.7}^{\beta=0.1}$

To have a better insight at the suggested robust performance, we plot in Figure 11 the evolution of the concentration ensemble members for all components in time. For a fair comparison, we also plot the associated reference solution, the EnKF's and the

EnKF-OI$_{\alpha=0.7}$'s ensemble members. As explained in section (3.4), the initial ensemble spread is clearly far from the truth. When data is assimilated into the system, all schemes tend to move closer to the truth. By the end of the 50-years period, both EnKF and EnKF-OI$_{\alpha=0.7}$ underestimate the concentration of DCE and VC and end up with quite small ensemble spread. The EnKF-OI$_{\alpha=0.7}^{\beta=0.1}$, on the other hand, retains the best performance, matching the reference solution for all components. Moreover, this hybrid scheme is shown to preserve the ensemble spread around the true final concentrations. In terms of the estimated

degradation rates, we show in Figure 12 the temporal change of MSE for each individual degradation rate as they result from the EnKF-OI$_{\alpha=0.1...0.9}^{\beta=0.0}$ (top panels) and EnKF-OI$_{\alpha=0.7}^{\beta=0.1...0.9}$ (bottom panels). For all rates, the EnKF-OI$_{\alpha=0.7}^{\beta=0.1...0.9}$ performs much better during the first 10 years, especially for $K_P$ and $K_T$. Averaging in time and over all cases, EnKF-OI$_{\alpha=0.7}^{\beta=0.1...0.9}$ is 33%, 17%, 33% and 15% more accurate for retrieving $K_P$, $K_T$, $K_D$ and $K_V$, respectively. From the images, one can see that the accuracy of the degradation rates tends to improve in time except for $K_D$ that is shown to degrade for small $\alpha$ and large

$\beta$ values. To interpret this behaviour, one should recall that $K_D$ (and also $K_V$) can only be estimated correctly as long as the concentration of the source component ($C_{\text{DCE}}$ in this case) is accurately recovered. Before attaining this, the estimates of $K_D$ are compensated for errors in $K_P$ and $K_T$.

Next, and instead of manually changing the weighting factors $\alpha$ and $\beta$, we follow eqs. (13) and (14) and conduct a 1D optimisation problem prior to assimilating the observations serially. The idea is to get the maximum reduction in the prior

uncertainties for both the concentration and the degradation rates. As such, different weights can be assigned to the background and the ensemble statistics. Based on this, we plot in Figure 13 the optimal $\alpha$ values at every assimilation cycle and for each observation. Recall that there is a total of 160 observations, such that each contaminant component is observed at 40 different locations. To better interpret the plot, we arrange these observation indices as follows: from top to bottom of the left y-axis; PCE: $1 \rightarrow 40$, TCE: $41 \rightarrow 80$, DCE: $81 \rightarrow 120$ and VC: $121 \rightarrow 160$. As can be seen from the plot, the adaptive EnKF-OI

algorithm selects either 0, and thus eq. (10a) is purely based on $\mathbf{P}_{xx}^b$, or 1 so that only the ensemble covariance, $\widehat{\mathbf{P}}_{xx}^{\text{EnKF}}$, is included. When assimilating PCE, TCE and VC concentrations, the adaptive scheme tends to use the background covariance (i.e., $\alpha = 0$) for almost the first 25 years. Beyond this, the filter statistics are only based on the ensemble flow-dependent information (i.e., $\alpha = 1$). This is not surprising given the large initial uncertainties imposed on the contaminant concentrations. Once the statistics are adjusted towards the truth, the filter starts trusting the information coming from the ensemble statistics.

This adaptive performance changes when assimilating DCE observations in the sense that the filter builds its forecast error





covariance mostly using background statistics and less using the flow-dependent ensemble. This comes in agreement with the analysis and the conclusion drawn from Figure 12 in which the background information of DCE are more useful than the ensemble statistics. Averaging over the entire optimal values of $\alpha$ values, we obtain a global $\alpha^* = 0.64$, which is quite close to the 0.7 value that resulted in the best performance in section (4.1.1). In terms of the adaptive $\beta$ values, we found out that

maximising the difference between the prior and the posterior parameters' covariance, $\widetilde{\mathbf{P}}_{\theta\theta}$, may not always be helpful. This is because doing such maximisation can quickly diminish the ensemble spread eventually paralysing the filter's analysis. In fact, minimising the difference[1] yielded more accurate degradation rates. To analyse this, we plot on the same figure the time-evolution of MSE of concentration when (1) maximising the information gain for both state and parameters, (2) minimising the information gain for both and (3) maximising the state's and minimising the parameters' information gain, respectively. As

demonstrated by the 3 curves, the best performance happens when the information gain for concentration is maximised and the associated parameter's one is minimised. Compared to maximising the information gain of both state and parameters, this mixed scheme now yields 37% more accurate contaminant concentrations.

## 4.2   The EnKF$_{ESOS}$ vs the EnKF

In the previous section, all approaches and experimental results were intended to overcome the rank deficiency and the under-

sampling of the ensemble's sample forecast error covariance. In the following experiment, we attempt to deal with the undersampling of the observation errors by implementing the EnKF$_{ESOS}$ algorithm presented in section (2.3). We first note that the distribution of the new observation perturbations show reasonable deviations from the prescribed Gaussian errors in the original EnKF algorithm, as has been noticed by Hoteit et al. (2015). To assess the performance of the EnKF$_{ESOS}$ against the EnKF, we study at a closer glance the contaminant maps after 50 years as estimated by the ensemble means from both schemes.

Thus, we plot in Figure 14 the normalised errors for the components TCE and DCE at layers 70 and 80, respectively. These error maps are obtained by subtracting the ensemble mean concentration from the reference and then normalising the result by the average of the reference solution. One common feature in these maps is the clear underestimation of TCE and DCE in the north part of the domain. This is because the initial reference concentration is quite different from the one assigned to the initial ensemble using the free run setup as outlined in section (3.4). In time, both filtering schemes try to push the contaminant

plume, which has already moved towards the southern region, upwards to match the truth. Moreover, as demonstrated in layer 70 and unlike the EnKF$_{ESOS}$, the EnKF overestimates TCE concentration in the center of the domain, which further continues to move southwards. In layer 80 (i.e., 5 m deeper), the EnKF tends to underestimate the concentration of DCE especially in the southern part of the domain. On the other hand, a slight overestimation of this DCE concentration towards the center is suggested by the EnKF$_{ESOS}$. In general, and assessing similar patterns at other layers, the EnKF$_{ESOS}$ shows higher accuracy in

retrieving the contaminant concentration than the EnKF. This provides further evidence that ignoring the observation sampling errors within the EnKF can indeed deteriorate the quality of the state estimates.

---

[1]Minimising the difference between the prior and the posterior covariances does not mean that the filter does not apply any correction. Since the Kalman analysis equation always minimises the variance, the adaptive algorithm now acts in a way such that only the lowest minimisation possible is retrieved. Unlike standard Kalman filtering, this procedure moves at a slower pace towards the truth.





To study the impact of the EnKF$_{ESOS}$ on the estimates of the parameters, we examine the evolution of the entire distribution of TCE degradation rate in time. We compare the resulting pdfs with those obtained using the EnKF after 5, 15, 30 and 45 years. On top of the pdfs, we also monitor the temporal evolution of $K_T$ AES in Figure 15. Starting from rather flat and

uncertain pdfs of $K_T$, both EnKF and EnKF$_{ESOS}$ exhibit a rigorous progress pushing the members of TCE degradation rate towards the truth, which is 0.086 per day. Notice that within the first 15 years, the pdfs seem to move in the wrong direction, away from the truth. This is due to the large concentrations at time 0, and thus the filter increases the degradation rates to fit the reference contaminant concentration. Beyond that and once the concentration is adjusted, the parameters from both filtering schemes begin moving closer to the true degradation rate. However, the EnKF is seen to move faster towards the truth and

further diminishes the uncertainty around $K_T$ quite rapidly. Consequently, the resulting pdf of $K_T$ after 45 years looks like a Kronecker delta function. This is, roughly speaking, not a very healthy assimilation system as the parameter updates become insignificant over the rest of the assimilation window. In contrast, the degradation rate obtained using EnKF$_{ESOS}$ moves at a slower pace towards the true rate maintaining large enough spread to fit the incoming observations. Compared to the EnKF, the AES suggested by the EnKF$_{ESOS}$, as shown on the left y-axis, is almost 40% to 50% higher. As a matter of fact, this

performance is more trustworthy than that of the EnKF indicating the essential need to account for observation sampling errors at the time of the analysis. Hoteit et al. (2015) found that the ensemble spread of the EnKF$_{ESOS}$ is larger than that of the EnKF for state estimation. In here, we experienced a similar, yet more pronounced behaviour for the estimates of the parameters.

As a final assessment, we compare the best estimates obtained using all considered schemes in this study; i.e., the EnKF, the EnKF-OI$_{\alpha=0.7}^{\beta=0.1}$ and the EnKF$_{ESOS}$. We plot the time series of MSE for contaminant concentration and degradation rates,

summed over all components, in Figure 16. Clearly the EnKF is the least accurate. Accounting for observation sampling errors yield around 21% and 23% more accurate state and parameters estimates, respectively. Tackling the rank deficiency of the EnKF results in 48% and 70% more accurate state and parameters estimates, respectively. Accordingly, addressing the issues of observation sampling errors and rank deficient forecast ensemble matrices seem to be crucial and can highly improve the accuracy of the estimates. From our experimental results and for this particular setting, resolving the rank deficiency issue

appear to have the largest impact on the final estimates of the filter.

## 5 Conclusions

In this study, we examined and investigated the hybrid ensemble Kalman filter (EnKF-OI) and the second-order observation perturbations sampling (EnKF$_{ESOS}$) schemes to estimate contaminant concentration and biodegradation rates of chlorinated hydrocarbons at the port of Rotterdam. We simulated the migration problem of a single plume consisting of Tetrachloroethene

(PCE), Trichloroethene (TCE), cis-1,2-Dichloroethene (DCE) and Vinyl Chloride (VC). Concentration data was used for yearly assimilation over a period of 50 years. The hybrid scheme complements the flow-dependent sample ensemble covariance of the EnKF with a prescribed static background covariance from an OI system to mitigate the undersampling of the ensembles and neglected model errors. The exact second-order sampling of the observation perturbations modifies the observation per-turbations and assimilates the data one after the other, thus resolving the undersampling of the observation noise in the EnKF





analysis. Challenging assimilation scenarios using a relatively small ensemble ($N_e = 48$) were presented, in which observations were processed serially. The key findings of this study and future research directions are summarised below:

1. Both the hybrid EnKF-OI and the EnKF$_{ESOS}$ successfully provide accurate concentration and degradation rate estimates. On average, a tuned hybrid EnKF-OI (using $\alpha = 0.7$ and $\beta = 0.1$) suggests 48% and 70% more accurate state and parameters estimates than those obtained using the EnKF. On the other hand, the EnKF$_{ESOS}$'s state and parameters estimates are 21% and 23% more accurate, respectively. In addition, the two schemes are easy to implement and computationally efficient requiring only a minimal change to the standard EnKF code.

2. Both filtering schemes demonstrated a better handling of the ensemble spread, for both state and parameters, avoiding any collapse or false (unrealistic) confidence in the estimates, leading to better ability to fit the observations.

3. The hybrid scheme requires some effort to tune two weighting factors that adjust the background statistics for both state and parameters. The serial adaptive version of this scheme, which relies on maximising the information gain between the forecast and analysis for each individual observation point, seems promising. From the experiments, we found that maximising the information gain however could possibly deplete the uncertainty within the ensemble quite rapidly. Yet, this observation may vary between systems depending on the degree and the rate of uncertainty growth. One possible solution that we tested is to minimise the information gain, and thus decrease the update impact when fitting the observations. Further, one could also build the objective function in such a way that only a portion of the information gain is maximised. For instance, an example would be to enforce the ratio between the trace of the analysis and the forecast covariance matrices to be greater than 40% meaning that at least 60% of the ensemble uncertainty is preserved.

4. Failing to account for observation undersampling errors in the standard EnKF can impact not only the quality of the state but more importantly the estimates of the parameters. In our experiments, the degradation rates obtained after assimilating using the EnKF$_{ESOS}$ scheme were more accurate, more reliable and certainly more realistic.

5. Careful tuning of the hybrid EnKF-OI yields the best estimates of the concentration and the degradation rates as compared to the EnKF and the EnKF$_{ESOS}$. This manifests the importance of hybridising the parameters cross-correlation matrix.

6. Building a unified EnKF scheme, which tackles both the undersampling of the forecast covariance and the observation sampling errors simultaneously is an interesting line of research in the future. The idea, as we see it now, would be to hybridise the joint ensemble statistics and then apply an SVD decomposition on the forecast perturbations. Thus instead of removing one rank only, $N_b + 1$ ranks need to be removed where $N_b$ is the rank of the background covariance. Subsequently, a second-order sampling of the observation perturbations could be then applied as described in section (2.3).



*Acknowledgements.*   The authors would like to thank the municipality of Rotterdam for their support and cooperation throughout this project. Research reported in this publication was supported by King Abdullah University of Science and Technology (KAUST).



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




## Table/Figure Captions:

**Table 1:** Different modelling parameters for the coupled flow-transport-reaction model. The values given for 3D parametric fields, such as bulk density and distribution coefficients, are the mean values from the entire 121 layers.

**Figure 1:** Schematic representation of the port of Rotterdam area with three main geologic layers: (i) Holocene clay and peat layer with sandy deposits ($\approx 20$ m thick), (ii) Pleistocene layer with coarse sand ($\approx 10$ m thick), and (iii) Pleistocene clay layer ($\approx 30$ m thick). POC1, POC2 and POC3 refer to different planes of compliance at the port site.

**Figure 2:** Initial configuration and geometry of the study area, located at the port of Rotterdam. The blue part is the domain area ($1.5$ km$^2$) of each layer and the yellow region is the plume of chlorinated hydrocarbon contaminants located in layer 60 at a depth of $22.5$ m below the mean sea level. The green triangles indicate the measurement locations collected from layers 30, 50, 70 and 90.

**Figure 3:** 2D spatial configuration of sorption (distribution coefficient) for trichloroethene (TCE) averaged over the first 10 layers of the domain.

**Figure 4:** Groundwater (GW) hydraulic head configuration from four different active layers in the domain. The largest water head is located in the center of the domain and is equal to $1.5$ m. The water head deceases in the southern part of the domain. The flow is computed using MODFLOW and plotted using iMOD's graphical interface utility.

**Figure 5:** Contaminant plume after 50 years for PCE, TCE, DCE and VC in layers 40, 60, 80 and 100, respectively. Vertically, the contaminant plume tend to move downwards towards the Pleistocene clay layers and the second aquifer. In the lateral direction, displacement of the plume happens southwards.

**Figure 6:** A sketch illustrating the procedure followed to construct the background statistics, $\mathbf{P}^b_{xx}$ and $\mathbf{P}^b_{\theta x}$. 3-months forecasts are performed starting from different initial conditions, $\mathbf{x}_{0,1,\ldots,N}$, and different degradations rate parameters, $\mathbf{\Theta}_{0,1,2\ldots,N}$, where $N = 200$ steps summing up to 50 years. The background state covariance, $\mathbf{P}^b_{xx}$, and state-parameters cross-correlations, $\mathbf{P}^b_{\theta x}$, are then constructed using the first leading modes only.

**Figure 7:** Individual cross-correlation terms of the background matrix $\mathbf{P}^b_{\theta x}$ associated with PCE biodegradation rate and VC concentration. The correlations are shown for all layers, assuming that the cells from each layer have been stretched in one vertical line (y-coordinate). Largest correlation is present in layer 60 where the contaminant source is located. Biodegradation in shallow layers is not as strong as in deep layers because of the downwards groundwater flow direction.

**Figure 8:** Left panel: Average mean square errors (MSE) of contaminant concentrations obtained using the EnKF and the hybrid EnKF-OI using $\alpha = 0.1, 0.2, \ldots, 0.9$. Right panel: Average MSE for PCE, TCE, DCE and VC biodegradation rates obtained using the EnKF and the hybrid EnKF-OI for different weight factor ($\alpha$) values.





**Figure 9:** Time series change of average ensemble spread (AES) resulting from the EnKF and hybrid EnKF-OI using 48 members in which $\alpha = 0.1, 0.2, \ldots, 0.9$.

**Figure 10:** Bar-Plot (left y-axis): The AES of concentration ensemble obtained using the hybrid EnKF-OI by changing the individual weighting factors ($\alpha$ and $\beta$) between 0.1 and 0.9. Shown according to the right y-axis is the MSE obtained using the hybrid EnKF-OI$_{\alpha=0.7}^{\beta=0.1,\ldots,0.9}$ (triangles), the hybrid EnKF-OI$_{\alpha=0.7}$ (cross) and the EnKF (plus).

**Figure 11:** Forecast ensemble members of PCE, TCE, DCE, and VC concentration. The evolution of these members ($N_e = 48$) is shown for the entire 50 years. Results are obtained using the standard EnKF, the EnKF-OI$_{\alpha=0.7}$ and the EnKF-OI$_{\alpha=0.7}^{\beta=0.1}$ schemes. Solid dashed lines correspond to the reference concentration of each component.

**Figure 12:** Images showing the MSE of PCE, TCE, DCE and VC degradation rates in time. These are obtained using the EnKF-OI scheme for (1) different $\alpha$ values (top panel) and (2) different $\beta$ values keeping $\alpha$ fixed and equal to 0.7
(bottom panel).

**Figure 13:** The coloured image shows, according to the left y-axis, the adaptive change in $\alpha$ values for each individual observation. The observation index $(1, \ldots, 160)$ is sorted such that the first 40 indices correspond to PCE measurements, the second 40 correspond to TCE, third 40 correspond to DCE and finally VC takes the last 40 indices. The yellow color indicates that no background covariance matrices have been used and the blue color suggests that only ensemble
"flow-dependent" statistics are involved. The curves demonstrate the change in MSE, according to the right y-axis, in time when maximising the information gain (cyan), minimising the information gain (red) and maximising the formation for concentration and minimising it for degradation rates (green).

**Figure 14:** Top panel: TCE concentration and error maps in layer 70 obtained using the reference run (1st column), the EnKF (2nd column) and the EnKF$_{ESOS}$ (3rd). Bottom panel: Same as top panel but for the concentration of DCE.

**Figure 15:** Left y-axis: The time evolution of the prior probability density functions corresponding to TCE degradation rate obtained using the EnKF (solid lines) and the EnKF$_{ESOS}$ (dashed lines). The reference "true" rate is given at 0.086 /day in brown color. Right y-axis: The AES of $K_T$ suggested using the EnKF and the EnKF$_{ESOS}$.

**Figure 16:** Left panel: Time-series of MSE for concentrations obtained using the EnKF, the EnKF$_{ESOS}$ and the hybrid EnKF-OI$_{\alpha=0.7}^{\beta=0.1}$. Right panel: Same as the left panel but for all degradation rates (y-axis is in log scale).




| Symbol | Parameter Description | Value (unit) |
|---|---|---|
| $\phi$ | Porosity | 0.30 |
| $\rho_b$ | Bulk density | 1167 (kg/m$^3$) |
| $k_{PCE}$ | Distribution coefficient of PCE | 0.0012 (m$^3$/kg) |
| $k_{TCE}$ | Distribution coefficient of TCE | 0.0015 (m$^3$/kg) |
| $k_{DCE}$ | Distribution coefficient of DCE | 0.0014 (m$^3$/kg) |
| $k_{VC}$ | Distribution coefficient of VC | 0.0010 (m$^3$/kg) |
| $\kappa_L$ | Longitudinal dispersivity | 0.5 (m) |
| $\dot{\kappa}_T/\kappa_L$ | Ratio of horizontal transverse dispersivity to longitudinal dispersivity | 0.1 |
| $\ddot{\kappa}_T/\kappa_L$ | Ratio of vertical transverse dispersivity to longitudinal dispersivity | 0.1 |
| $\delta_m$ | Molecular diffusion | $10^{-10}$ (m$^2$/s) |





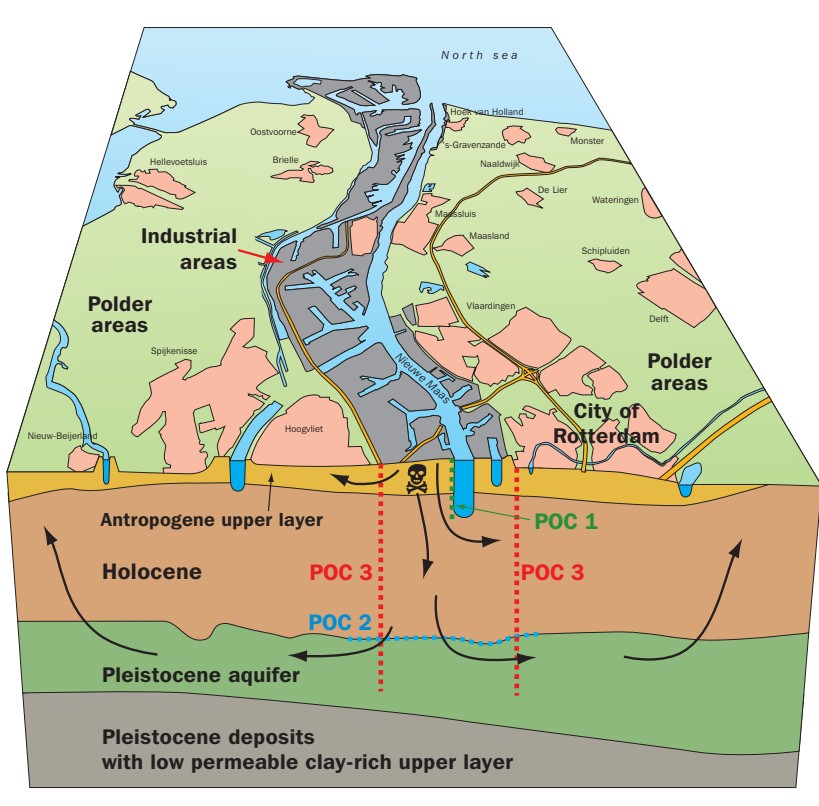


























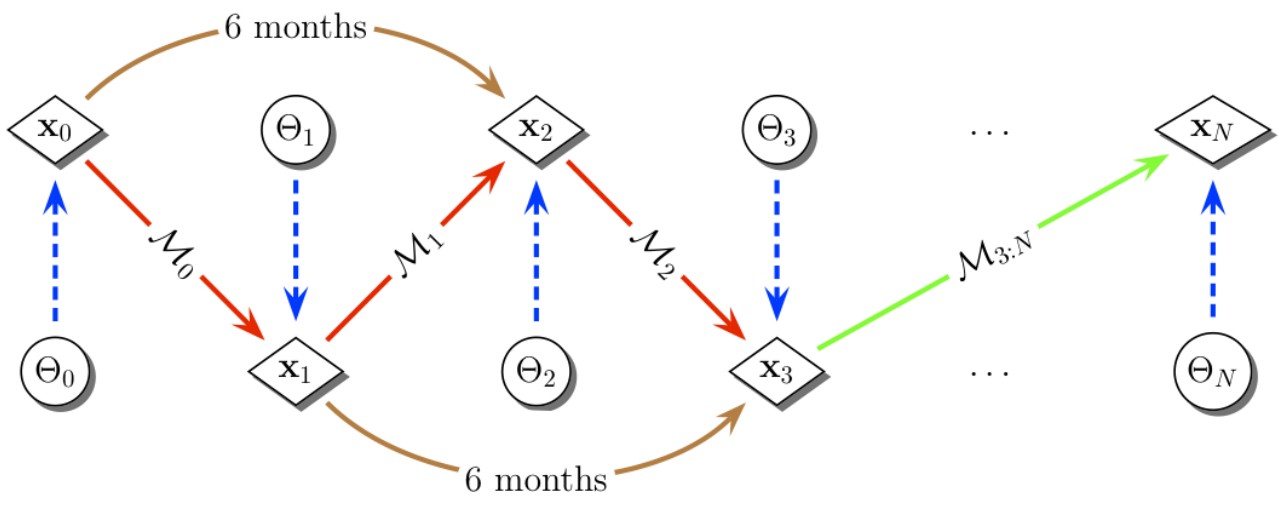









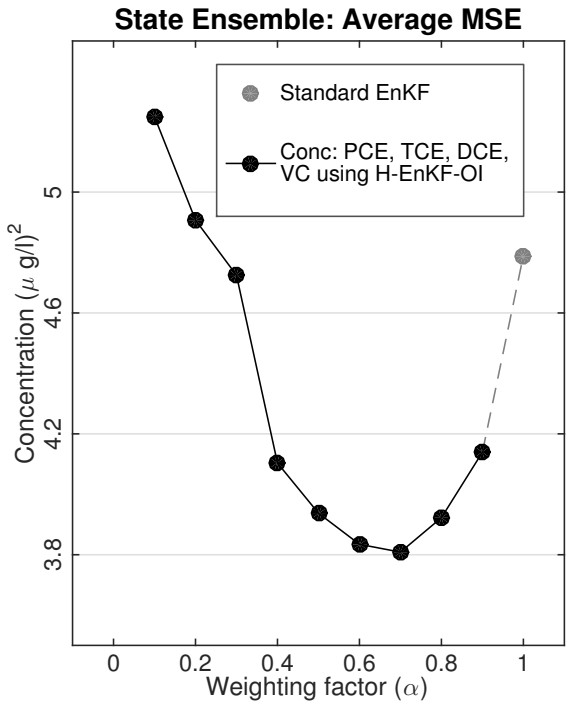

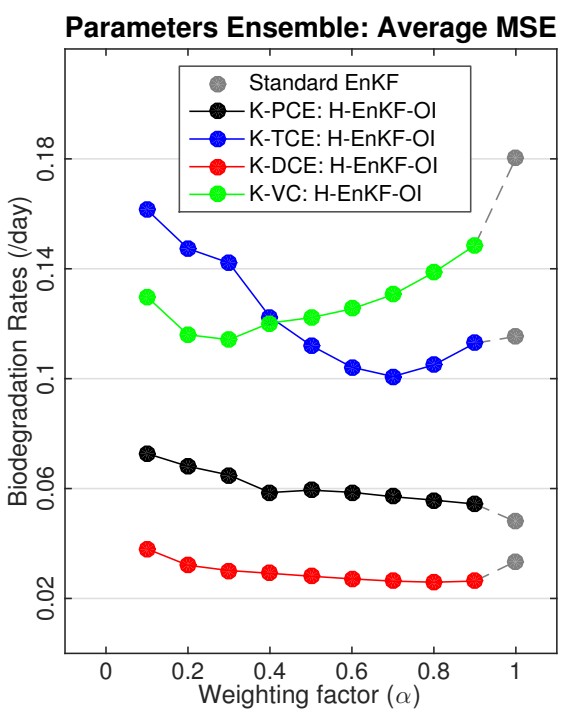

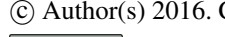
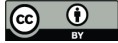

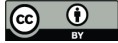

**Parameter Statistics: Average-Ensemble-Spread**

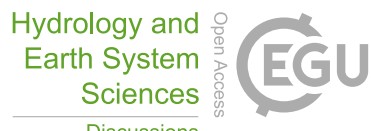



**Concentration Ensemble Statistics**






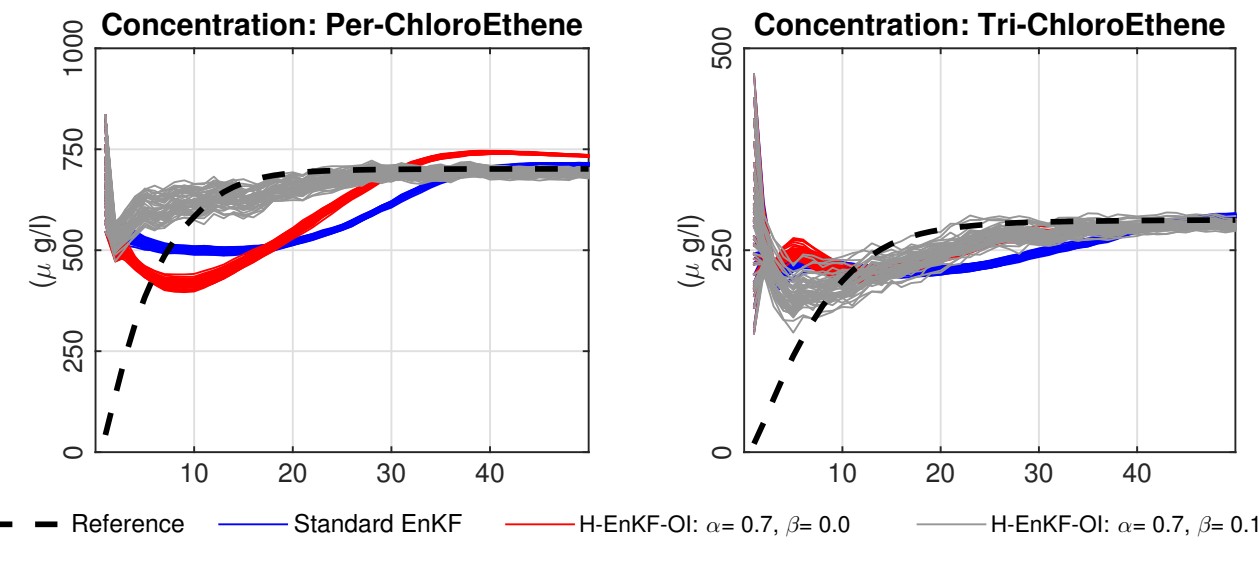

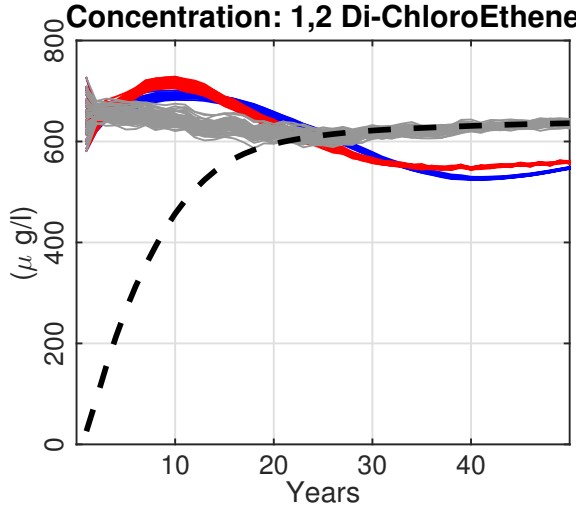

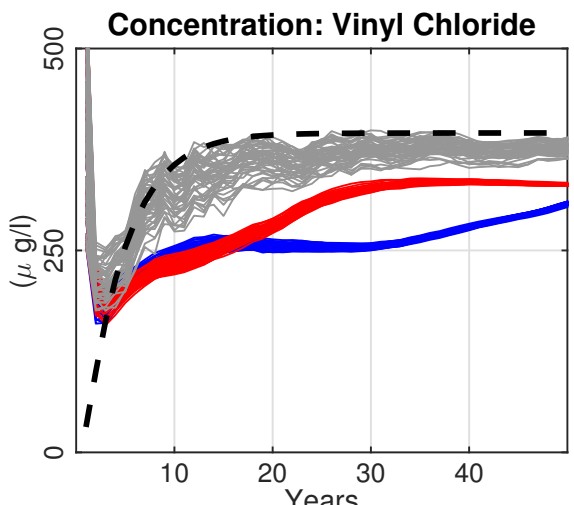





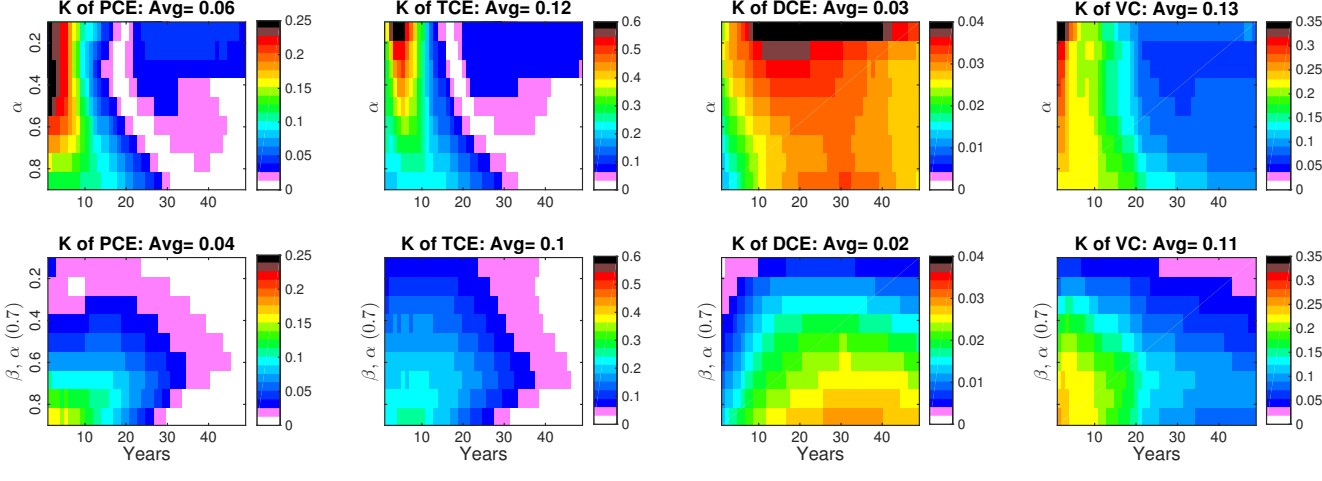







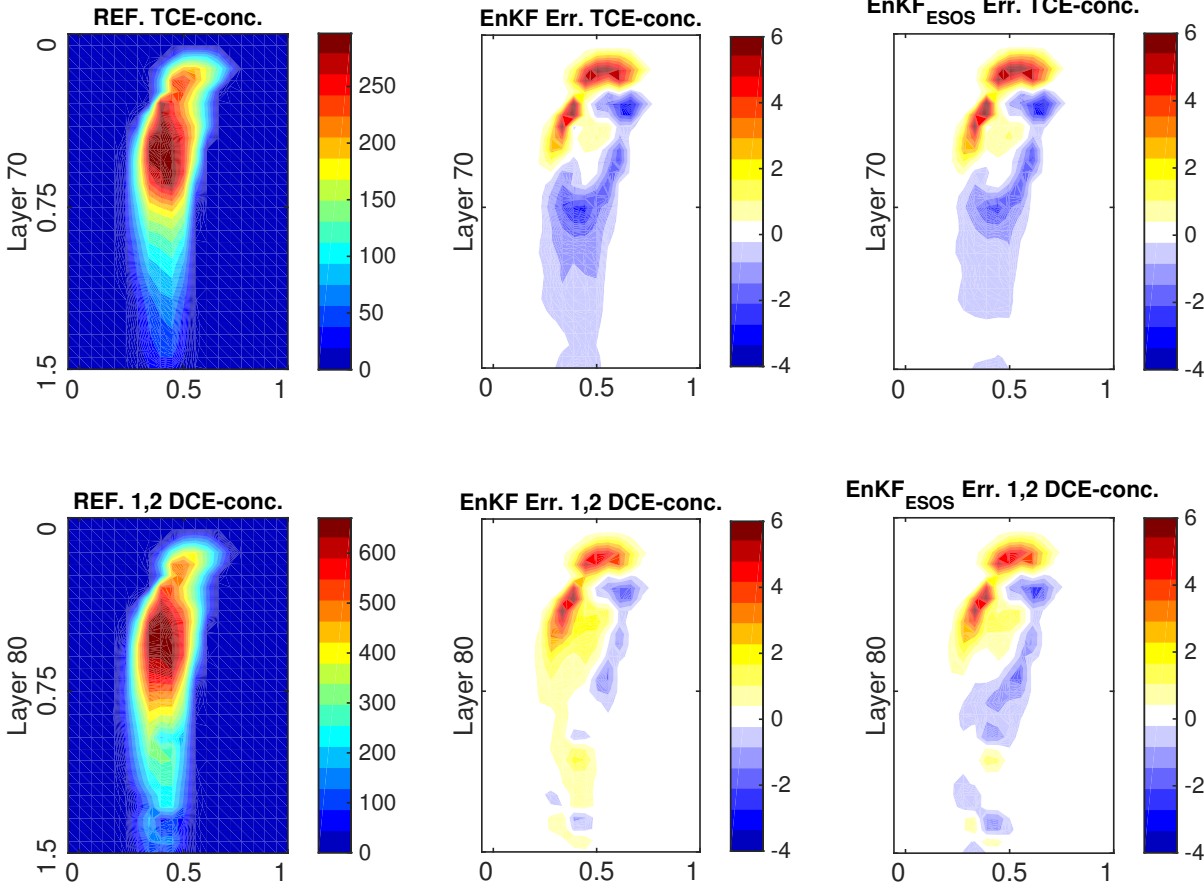





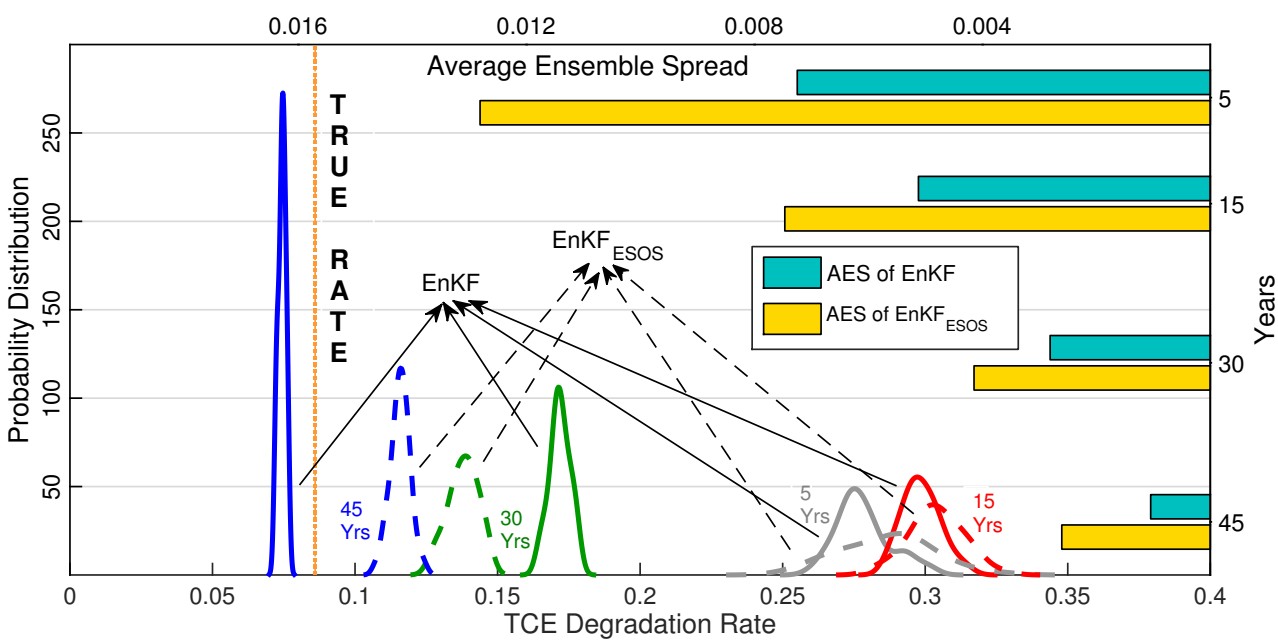






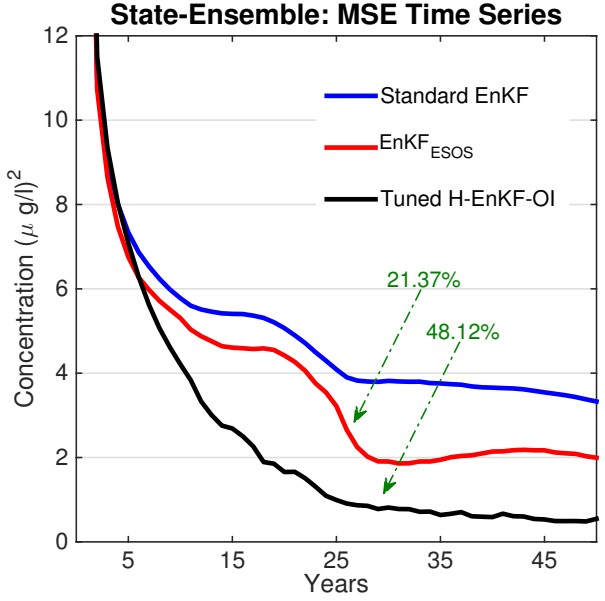
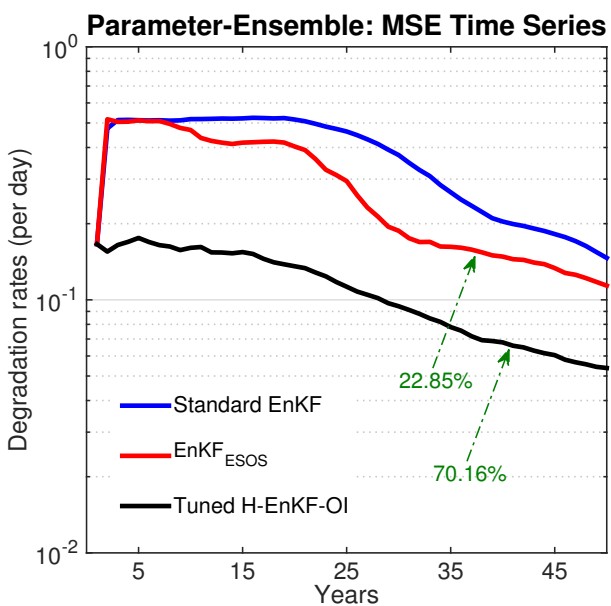