# Peer review of "On the efficiency of the hybrid and the exact second-order sampling formulations of the EnKF: A reality-inspired 3D test case for estimating biodegradation rates of chlorinated hydrocarbons at the port of Rotterdam"

_Hydrology and Earth System Sciences, 2016_

## Referee Comment (RC1) · Anonymous Referee #1 · 11 May 2016

In this case the title says it all, or almost... I was quite thrilled when I read the title and introduction since I was expecting to see an application of the EnKF to a realistic case (port-Rotterdam inspired), unfortunately the final outcome is a nicely written, quite interesting analysis of the efficiency of the hybrid and the exact second-order sampling formulations of the EnKF, but the application, although port-Rotterdam-inspired, is far from being realistic at all. And the authors fail to recognize it.

The authors make no comment about the statement in line 345ff "Modelling parameters

required for running the coupled FTR-Model, such as porosity, distribution coefficients and others are defined, based on real data and laboratory assessment, as 3D heterogeneous fields" (They forgot to mention explicitly hydraulic conductivity.)

This assumption means that all the uncertainty associated with the heterogeneous geological parameters is discarded, and that all the analysis has been performed assuming that porosity, conductivity, distribution coefficient, and other parameters are perfectly known. Once this is realized, one has to continue reading under the understanding that what follows is a purely academic exercise, poorly disguised as a realistic application.

The authors must be very clear from the very beginning on this "small" detail, and acknowledge it. Apart from that, I think the paper is well written, hard to follow at times, and provides an interesting discussion on how to deal with the specifics of the hybrid and the exact second-order sampling formulations of the EnKF.

Minor comments

line 129: What do you mean by "...the EnKF computes an approximation of the joint pdf..." Unless you mean the non-parametric joint pdf as implied by the raw set of ensemble values, the statement is incorrect. The EnKF is based on means and covariances, but this does not imply that by knowing them you know the joint pdf.

line 160. There is no Gaussian assumption in the derivation of the Kalman filter equations!! Those equations are solely based on means and covariances and there is no requirement that parameters or state variables are Gaussian to derive them. However, it is true that the EnKF is optimal for multiGaussian-based variables.

line 483. ... the "famous" steady-state Kalman filter... Please, watch your wording and avoid sensationalism.

---

## Referee Comment (RC2) · Anonymous Referee #2 · 20 May 2016

Review of the paper titled "On the efficiency of the hybrid and the exact second-order sampling formulations of the EnKF: A reality-inspired 3D test case for estimating biodegradation rates of chlorinated hydrocarbons at the port of Rotterdam" submitted to Hydrol. Earth Syst. Sci.

The paper presents an assimilation approach to subsurface contaminant transport problem inspired by the port of Rotterdam in the Netherlands. A multi-dimensional and multi-species reactive transport model is coupled to simulate the migration of contami-

nants within subsurface flow model. The biodegradation chain of chemicals is modelled for five decades. An artificial measurement data for the concentration is build using a synthetic setup and then used for updating the concentration and degradation rates in presence of model and observational errors. An adaptive hybrid ensemble Kalman filter is evaluated along side the exact second-order sampling formulation introduced by one of the authors in an earlier publication. The paper is well written and the presented numerical results are interesting. However, the test setup assumed perfect knowledge of the distributed subsurface parameters (permeability and porosity) which is generally unknown except at few locations.

I find the results convincing but would recommend that the authors add the following to the numerical study:

1- The utilized models and state parameter estimation techniques are limited to online updating systems which in many cases are known to under-perform iterative schemes (ensemble smoother where all the data is assimilated at once) specially within an annealing framework in what is known as ensemble smoother with multiple data assimilation. Can the author include that in their numerical study.

2- Could the authors re-run the model with the estimated parameters from the initial time step without data assimilation to assess the quality of the estimated parameters.

3- Uncertainties in spatial parameters (permeability and porosity) is a very interesting topic. Can the authors include some elements of that in their study even in a simplified way?

---

## Referee Comment (RC3) · Anonymous Referee #3 · 21 May 2016

In the paper by Gharamti et al., the authors compare three data assimilation strategies for a subsurface state-parameter estimation problem: the standard ensemble Kalman filter (EnKF), a hybrid EnKF including optimal interpolation (EnKF-OI) and a second-order sampling formulation of EnKF (EnKF-ESOS). Synthetic data assimilation experiments are performed with a reactive transport problem for migration, sorption an degradation of chlorinated hydrocarbons. This set-up should mimic a contaminated aquifer in the port of Rotterdam. Concentration data and first-order degradation rates are updated within the three assimilation schemes.

The paper is well written and points out important limitations of the ensemble Kalman filter in subsurface characterization (undersampling of forecast covariances and observation errors) and how they could be ameliorated with EnKF-IO and EnKF-ESOS. However, I have two major concerns regarding the content of the paper:

(I) It seems to me that there is a considerable overlap with earlier work from Gharamti et al. (2014). Large parts of the paper related to the EnKF-OI contain very similar information as in the earlier work and also the overall model set-up is quite similar in both studies (see below) leading to almost the same conclusions regarding EnKF-OI. Therefore, the authors should give a clear motivation why the comparison EnKF/EnKF-OI is repeated in this paper and they should point out what is the innovative aspect of this study compared to their previous work (i.e., what did we learn from this study regarding EnKF/EnKF-OI that was not already covered in Gharamti et al., 2014).

(II) The authors claim to use a 'reality-inspired' test case for the comparison of the different data assimilation schemes. In fact, only a limited amount of information about the site characterization is given in section 3.1 which makes it difficult for the reader to judge how realistic the model set-up is. For example, how many measurements were available to derive the parameter fields for hydraulic conductivity, porosity and distribution coefficients and how uncertain are the derived parameter fields? Is the model discretization fine enough to account for the spatial variability of subsurface parameters? Another question is whether the assumption of steady state groundwater flow is valid for the chosen site. Usually, one would expect transient groundwater flow due to temporally variable recharge, pumping activities or density-driven flow in such environments. Transient groundwater flow could have important implications for the data assimilation, e.g., for the determination of the background covariance matrix in the EnKF-OI scheme (see below). Overall, the current set-up is very similar to what has been used in Gharamti et al. (2014) except that groundwater flow is 3D in this example (which should not be a major issue when a steady state flow field is used) and

the chemical reactions are different (but follow a very similar mathematical description). So in fact, I think that there is not much more complexity in this 'reality inspired' set-up than in the 'purely' synthetic set-up used in previous studies. Therefore, I suggest the authors to add more complexity in their model set-up in order to test the different assimilation schemes under more realistic conditions. This could be accomplished e.g., by considering more sources of uncertainty (e.g. hydraulic parameters, forcing terms) and by using transient flow conditions.

Specific comments

Line 191-192: The same applies for the alpha and beta values in EnKF-OI.

Line 195-196: What do you mean with '...dynamically constants quantities....'?

Line 216-217: Incomplete sentence.

Line 368-370 and Figure 5: Why does PCE appear in layer 40, when the contaminant source is located in layer 60 and the pre-dominant flow direction is downward? Is the groundwater flow rate so low compared to molecular diffusion?

Line 396-417: In this example, the background covariance matrix for EnKF-OI is derived on the basis of a steady-state flow field with perfectly known hydraulic parameters. Additionally, the background covariances are derived from the same time period, where the assimilation experiments are performed. This means that the derived background covariance matrix contains a very precise description of the relation between concentrations and degradation rates in your system. However, under real-world conditions the uncertainties in hydraulic parameters may have a considerable impact on the quality of the background covariance matrix. Additionally, under transient flow conditions it might be much more difficult to derive a good estimate of the background covariance matrix. Therefore, I suggest the authors to discuss such practical issues in more detail and also to perform additional simulation experiments where these influences on the derivation of the background covariance matrix are assessed in more detail, e.g. by

introducing uncertainty in the hydraulic parameters and by using transient flow conditions. This would provide a more realistic assessment of the EnKF-OI assimilation scheme.

Figure 11: It would be helpful in this plot to also show the evolution of concentration values without data assimilation as a comparison. Additionally, why does the optimized EnKF-OI simulation (grey lines) for PCE update in the wrong direction between year 5 and 10?

References

Gharamti, M., Valstar, J., Hoheit, I. (2014): An adaptive hybrid EnKF-OI scheme for efficient state-parameter estimation for reactive contaminant transport models, Advances in Water Resources, 71, 1-15.

---

## Author Comment (AC1) · 20 Jul 2016

**Reply to Reviewer #1**

We would like to thank the reviewer for his comments and suggestions. Below please find our detailed response to the reviewer's concerns.

In this case the title says it all, or almost... I was quite thrilled when I read the title and introduction since I was expecting to see an application of the EnKF to a realistic case (port-Rotterdam inspired), unfortunately the final outcome is a nicely written, quite interesting analysis of the efficiency of the hybrid and the exact second-order sampling formulations of the EnKF, but the application, although port-Rotterdam-inspired, is far from being realistic at all. And the authors fail to recognize it.

The authors make no comment about the statement in line 345 "Modelling parameters required for running the coupled FTR-Model, such as porosity, distribution coefficients and others are defined, based on real data and laboratory assessment, as 3D heterogeneous fields" (They forgot to mention explicitly hydraulic conductivity.)

This assumption means that all the uncertainty associated with the heterogeneous geological parameters is discarded, and that all the analysis has been performed assuming that porosity, conductivity, distribution coefficient, and other parameters are perfectly known. Once this is realized, one has to continue reading under the understanding that what follows is a purely academic exercise, poorly disguised as a realistic application.

The authors must be very clear from the very beginning on this "small" detail, and acknowledge it. Apart from that, I think the paper is well written, hard to follow at times, and provides an interesting discussion on how to deal with the specifics of the hybrid and the exact second-order sampling formulations of the EnKF.

The reviewer raises a good point. Improving the estimates of the groundwater flow, on top of the contaminant dynamics (transport & reactions), is rather important. This is usually done, as the reviewer points out, by quantifying the uncertainties of the hydraulic parameters such as conductivity and porosity. This has been extensively studied in the Hydrology literature.

Our focus in this paper, however, is to address two major drawbacks of the EnKF; namely the forecast under-sampling and observation sampling errors. We present this while focusing on a slightly different, but related, application and that is quantifying the uncertainties associated with subsurface biodegradation reactions. To the best of our knowledge, this would be the first application in which biodegradation parameters are estimated in a near-realistic modeling scenario using the EnKF. Addressing the uncertainties of subsurface hydraulic parameters is possible but is beyond the scope of the current study. Following the reviewer's suggestion, we now clarify this detail in the introduction section. The reviewer may refer to lines 99-101.

Concerning line 345, we now provide more details on the offline procedure we follow to estimate the hydraulic properties of the subsurface such as porosity and conductivity. In essence, the hydraulic conductivity is provided in the database GeoTOP. The GeoTOP for the province of South-Holland is constructed using 46.000 borehole data. Using the borehole data, the most probable lithostratigraphy and lithofacies have been estimated in each voxel of 100x100x0.5 m. The GeoTOP further uses relations between the lithostratigraphical units and the lithofacies with parameters such as hydraulic conductivity, porosity and organic carbon content in order to provide these parameters on the voxel scale. Further details and essential referencing are provided towards the end of Section **3.2.2**.

**Minor comments**

Line 129: What do you mean by "...the EnKF computes an approximation of the joint pdf..." Unless you mean the non-parametric joint pdf as implied by the raw set of ensemble values, the statement is incorrect. The EnKF is based on means and covariances, but this does not imply that by knowing them you know the joint pdf.

Given the limited ensemble size, we refer to the joint pdf suggested by the EnKF at every forecast step as an approximation of the "true" pdf. We agree with the reviewer, having the mean and the covariance does not necessarily give us access to the entire true distribution of both state and parameters. This was made clearer in the revised text.

Line 160. There is no Gaussian assumption in the derivation of the Kalman filter equations!! Those equations are solely based on means and covariances and there is no requirement that parameters or state variables are Gaussian to derive them. However, it is true that the EnKF is optimal for multiGaussian-based variables.

By construction, the Kalman Filter accounts only for the first and second moments of the estimated random variable (state or parameters). When the pdfs are not Gaussian, it is only optimal along linear estimators and when both the model and observational operators are linear. As the reviewer suggests, when the distribution of the unknowns is multiGaussian (and the model is linear) the EnKF is optimal (for an infinite ensemble size). This is however almost never the case when parameters are also part of the state vector, as in the case of our study. The sentence was revised to remove any confusion.

Line 483. ... the "famous" steady-state Kalman filter... Please, watch your wording and avoid sensationalism.

The word famous has been removed. Thank you.

---

## Author Comment (AC2) · 20 Jul 2016

**Reply to Reviewer #2**

We thank the reviewer for commenting on the manuscript. Below please find our detailed response to the reviewer's concerns.

The paper presents an assimilation approach to subsurface contaminant transport problem inspired by the port of Rotterdam in the Netherlands. A multi-dimensional and multi-species reactive transport model is coupled to simulate the migration of contaminants within subsurface flow model. The biodegradation chain of chemicals is modeled for five decades. An artificial measurement data for the concentration is build using a synthetic setup and then used for updating the concentration and degradation rates in presence of model and observational errors. An adaptive hybrid ensemble Kalman filter is evaluated along side the exact second-order sampling formulation introduced by one of the authors in an earlier publication. The paper is well written and the presented numerical results are interesting. However, the test setup assumed perfect knowledge of the distributed subsurface parameters (permeability and porosity), which is generally unknown except at few locations.

We thank the reviewer for his positive feedback. Concerning the spatial distribution of the permeability (hydraulic conductivity) and porosity, we agree that quantifying their uncertainties is essential; however, it is beyond the scope of our work. In the revised manuscript, we now provide details on the offline estimation procedure that lead to a 3D parameterization of the flow parameters. Please refer to Section **3.2.2**.

I find the results convincing but would recommend that the authors add the following to the numerical study:

1- The utilized models and state parameter estimation techniques are limited to online updating systems which in many cases are known to under-perform iterative schemes (ensemble smoother where all the data is assimilated at once) specially within an annealing framework in what is known as ensemble smoother with multiple data assimilation. Can the author include that in their numerical study.

The objective of the current work is to test the usefulness of accounting for [1] EnKF forecast under-sampling issues (forecast step) and [2] EnKF observation sampling errors (analysis step). We then draw conclusions on how each of these two issues affects the accuracy and reliability of the resulting state and parameters' estimates. We realize that iterative ensemble schemes are convenient to apply in subsurface applications, but this is not the focus of our study. An ensemble smoother (ES) can still suffer from under-sampling issues during the forecast step because of the limited ensemble size. Adding the MDA scheme to the analysis step may help to improve the fit to the data when all are assimilated at once. It is further computationally more demanding and may suffer from convergence issues. In general, it can be subject to the same problems related to under-sampling of the background and observational error covariances as the EnKF. This study only considers the filtering problem. The smoothing framework could be considered in future studies. Thank you.

2- Could the authors re-run the model with the estimated parameters from the initial time step without data assimilation to assess the quality of the estimated parameters.

The reviewer is raising a good point here. Rerunning the simulation from the initial time using the estimated parameters is useful in real experiments in which the true

parameters are not known. In our twin-experimental setup, the true parameters are known (given by Suarez and Rifai, 1999) and thus the quality of the estimated parameters is directly assessed by how far are those from the true ones. This has been analyzed in figures 8, 12, 15 and 16. Nevertheless, we followed the reviewer's suggestion and we did a forward model run using the estimated parameter. We compared, please see Fig. 1 below, the resulting MSE for concentration to that obtained using the initial parameters (initial ensemble mean from the EnKF runs). As can be seen, the estimates of the concentration improve when using the estimated biodegradation parameters in the FTR-Model. Overall, the gain in concentration accuracy is about 24%. We will be happy to add this figure in the revised manuscript if the reviewer still thinks that it can be useful. Thank you.

[Figure]

**Figure 1: Free model run using the initial parameters and those estimated by the tuned hybrid EnKF-OI.**

3- Uncertainties in spatial parameters (permeability and porosity) is a very interesting topic. Can the authors include some elements of that in their study even in a simplified way?

Following the reviewer's suggestion, we added a Section (**4.3**) in which we analyze a new set of results based on perturbed flow hydraulic parameters (conductivity and porosity). An ensemble of these hydraulic parameters is created and used to run the coupled FTR-model. We found that imposing large uncertainties on the hydraulic parameters strongly degrades the performance of all filtering schemes. Given that the performance of the hybrid EnKF-OI depends on the quality of the background statistics, satisfactory results were obtained only when the uncertainty imposed on the background information is relatively moderate. Further details can be found in the revised manuscript. Thank you.

---

## Author Comment (AC3) · 20 Jul 2016

**Reply to Reviewer #3**

We would like to thank the reviewer for his thorough review. We appreciate his time and effort and all his suggested comments, which improved the quality of our work. We followed the reviewer's suggestions and revised the manuscript accordingly. Below please find our detailed response to the reviewer's comments.

In the paper by Gharamti et al., the authors compare three data assimilation strategies for a subsurface state-parameter estimation problem: the standard ensemble Kalman filter (EnKF), a hybrid EnKF including optimal interpolation (EnKF-OI) and a second order sampling formulation of EnKF (EnKF-ESOS). Synthetic data assimilation experiments are performed with a reactive transport problem for migration, sorption and degradation of chlorinated hydrocarbons. This set-up should mimic a contaminated aquifer in the port of Rotterdam. Concentration data and first-order degradation rates are updated within the three assimilation schemes.

The paper is well written and points out important limitations of the ensemble Kalman filter in subsurface characterization (under-sampling of forecast covariances and observation errors) and how they could be ameliorated with EnKF-IO and EnKF-ESOS. However, I have two major concerns regarding the content of the paper:

1- It seems to me that there is a considerable overlap with earlier work from Gharamti et al. (2014). Large parts of the paper related to the EnKF-OI contain very similar information as in the earlier work and also the overall model set-up is quite similar in both studies (see below) leading to almost the same conclusions regarding EnKF-OI. Therefore, the authors should give a clear motivation why the comparison EnKF/EnKF-OI is repeated in this paper and they should point out what is the innovative aspect of this study compared to their previous work (i.e., what did we learn from this study regarding EnKF/EnKF-OI that was not already covered in Gharamti et al., 2014).

We thank the reviewer for pointing this out. The previous work only introduced the hybrid EnKF-OI formulation to state-parameters estimation problems. We agree that part of the methodology has some overlaps but the overall goal of the two studies is quite different. The major differences between both studies are listed here:

   a. The update step of the hybrid EnKF-OI algorithm is extended. We allow the observations to be processed serially, and therefore the optimization presented in *Gharamti et al. (2014)* to be performed for each single observation separately. We believe that this is a more convenient approach given that different observations carry varying degrees of information to the system. As such the weighting between the ensemble and the background covariances changes when assimilating the observations serially. The serial assimilation makes the update scheme consistent with that of the EnKF-ESOS, which requires the observations to be processed serially.

   b. The application presented in this study is based on a large-scale and more realistic problem. Although the reviewer seems to think that the model setup is similar, there are many differences to the one used in *Gharamti et al. (2014)*. In particular:
      i.   The model is three-dimensional (unlike the 2D problem in the previous article) and the hydraulic parameters such as porosity and permeability

are based on real geologic facies. In the revised manuscript, we now outline the procedure we follow to construct the parameters (using the GeoTOP software package). The masking of the domain location (in the port area) and the confidentiality of the contaminant data are two conditions imposed by the municipality of Rotterdam (it is not in our control). Further details on the flow model parameters are now included in Section **3.2.2**.

    ii.   The vertical resolution of the model is fine and quite unique compared to many other model setups found in the literature. Many studies assume a single layer (maybe 2 or 3 at most) for each aquifer. We however discretize the vertical model domain into 120 layers, covering 4 aquifer systems, each of length 0.5 m. This helps to understand the interaction between the components and eventually provide more insights on the correlations between the parameters and the associated component concentrations.

c.  Different optimization strategies for determining the weighting factors in the hybrid algorithm are now examined. *Gharamti et al. (2014)* only considered maximizing the information gain to weight between the flow-dependent and the static covariances. In the current study, we test and analyze different optimization scenarios (Section **4.1.2**). For instance, maximizing the information gain when hybridizing the state statistics (i.e., $\alpha$) and minimizing information gain when hybridizing the parameters statistics (i.e., $\beta$). We also assess the performance when the information gain is minimized for both state and parameters statistics (refer to lines 71-73 and 543-548).

d.  The EnKF-ESOS algorithm is not yet tested for state-parameters estimation problems. It was only presented for state estimation only (refer to lines 85-87).

e.  One important message we emphasize in this manuscript is the efficiency of each approach (OI and ESOS) within the EnKF. In other words, we quantify the improvements that could be achieved when we tackle the under-sampling issues that are related to the limited ensemble size or the observational sampling errors. This is discussed in Section **4.2** (and lines 621-626).

2- The authors claim to use a 'reality-inspired' test case for the comparison of the different data assimilation schemes. In fact, only a limited amount of information about the site characterization is given in section 3.1, which makes it difficult for the reader to judge how realistic the model set-up is. For example, how many measurements were available to derive the parameter fields for hydraulic conductivity, porosity and distribution coefficients and how uncertain are the derived parameter fields? Is the model discretization fine enough to account for the spatial variability of subsurface parameters? Another question is whether the assumption of steady state groundwater flow is valid for the chosen site. Usually, one would expect transient groundwater flow due to temporally variable recharge, pumping activities or density-driven flow in such environments. Transient groundwater flow could have important implications for the data assimilation, e.g., for the determination of the background covariance matrix in the EnKF-OI scheme (see below). Overall, the current set-up is very similar to what has been used in Gharamti et al. (2014) except that groundwater flow is 3D in this example (which should not be a major issue when a steady state flow field is used) and the chemical reactions are different (but follow a very similar mathematical description).

So in fact, I think that there is not much more complexity in this 'reality inspired' setup than in the 'purely' synthetic set-up used in previous studies. Therefore, I suggest the authors to

add more complexity in their model set-up in order to test the different assimilation schemes under more realistic conditions. This could be accomplished e.g., by considering more sources of uncertainty (e.g. hydraulic parameters, forcing terms) and by using transient flow conditions.

We thank the reviewer for bringing this discussion about the model. We now discuss the initialization process for the parameters and further study the impact of model uncertainty on the performance of the schemes. Our response for each point is detailed below:

a.  The hydraulic conductivity is provided in the database GeoTOP. The GeoTOP for the province of South-Holland is constructed using 46.000 borehole data. Using the borehole data, the most probable lithostratigraphy and lithofacies have been estimated in each voxel of 100x100x0.5 m. In the next step the GeoTOP uses relations between the lithostratigraphical units and the lithofacies with parameters such as hydraulic conductivity, porosity and organic carbon content in order to provide these parameters on the voxel scale. Further details and related references are now included in Section **3.2.2** of the revised manuscript.

b.  The horizontal model discretization (50 m) is finer than the resolution of the parameters such as the hydraulic conductivity (100 m) and the vertical dimensions are equal. The vertical discretization is 0.5 m (for each layer) and this is a considerably fine resolution.

c.  In our opinion, steady state groundwater flow is a valid assumption. We agree that there are temporal variations on a small scale, such as tidal influences and yearly fluctuations of precipitation and evapotranspiration. Effects of tidal influences are expected to be minor (yearly averaged additional advection is zero but it may increase spreading that is accounted for by a relatively high effective dispersivity values). Effects of yearly fluctuations of precipitation and evapotranspiration are also expected to be minor as the near surface groundwater levels are controlled by the drainage levels of the drainage systems in the port area (3 – 4 m above sea level) and the deeper groundwater levels are predominantly influenced by the surface water levels in both the polders area  (managed levels around or below sea level) and the large surface waters (approximately sea level). Temporal variations due to density driven flow can be also neglected as we would expect only minor changes in the most lower part of the model domain on the time scale of 50 years. Similar discussion has been added to Section **3.2.2**.

d.  The reviewer is raising an interesting point regarding the background covariance of the hybrid EnKF-OI and the connection to perfect flow conditions. Following the reviewer's suggestion, we have added a new section (**4.3**) to the results in which we run a new set of experiments while perturbing the hydraulic conductivity and the porosity. We test the performance when strong and moderate uncertainties are imposed. We found that imposing large uncertainties on the hydraulic parameters strongly degrades the performance of all filtering schemes. Given that the performance of the hybrid EnKF-OI depends on the quality of the background statistics, satisfactory results were obtained only when the uncertainty imposed on the background information is moderate and not very high. Further details can be found in the Abstract,

Section **4.3** and Figures 18 and 19 of the revised manuscript. Thank you.

**Specific comments**

Line 191-192: The same applies for the alpha and beta values in EnKF-OI.

Yes, if they are set manually. Our proposed adaptive scheme does not require any tuning effort.

Line 195-196: What do you mean with '...dynamically constants quantities....'?

We meant to say that they are static in time unlike the state (e.g., concentration) of model, which evolves based on the dynamics of subsurface. We clarify this in the revised manuscript.

Line 216-217: Incomplete sentence.

We modified the sentence, which reads now: "This decomposition is useful in practice in order to reduce computational burden and memory storage." Thank you.

Line 368-370 and Figure 5: Why does PCE appear in layer 40, when the contaminant source is located in layer 60 and the pre-dominant flow direction is downward? Is the groundwater flow rate so low compared to molecular diffusion?

Overall, the groundwater flow rate (in the downward direction) is stronger than molecular diffusion. However, since we consider a constant source term for PCE in layer 60 and given the long simulation time (i.e., 50 years) a small amount of this component appear in layer 40 (under the effect of molecular diffusion). We checked the other 3-contaminant components, and none of them reach layer 40 by the end of the simulation period. Thank you.

Line 396-417: In this example, the background covariance matrix for EnKF-OI is derived on the basis of a steady-state flow field with perfectly known hydraulic parameters. Additionally, the background covariances are derived from the same time period, where the assimilation experiments are performed. This means that the derived background covariance matrix contains a very precise description of the relation between concentrations and degradation rates in your system. However, under real-world conditions the uncertainties in hydraulic parameters may have a considerable impact on the quality of the background covariance matrix. Additionally, under transient flow conditions it might be much more difficult to derive a good estimate of the background covariance matrix. Therefore, I suggest the authors to discuss such practical issues in more detail and also to perform additional simulation experiments where these influences on the derivation of the background covariance matrix are assessed in more detail, e.g. by introducing uncertainty in the hydraulic parameters and by using transient flow conditions. This would provide a more realistic assessment of the EnKF-OI assimilation scheme.

The reviewer has a good point. As mentioned in our response to the reviewer's second major comment, we now include a set of experiments that are based on imperfect hydraulic parameters. In Figure 17, we analyze the effect of perturbing the flow model on the background cross-correlations. Essentially, we found that the dominant correlation patterns are similar to those obtained using perfect flow conditions, especially in the shallow aquifer layers. The magnitude of the new background correlations, however, is considerably smaller. Generally, porosity and conductivity affect the speed and the movement of groundwater in the aquifer and thus the

degradation process would be expected to either slow down or accelerate. A similar discussion has been included in the revised manuscript (lines 602-613). Thank you.

Figure 11: It would be helpful in this plot to also show the evolution of concentration values without data assimilation as a comparison. Additionally, why does the optimized EnKF-OI simulation (grey lines) for PCE update in the wrong direction between year 5 and 10?

We have added the free run concentration evolution to Figure 11. The update of PCE between year 5 and 10 does not entirely go in the wrong direction. Over all, the concentration is close to the reference solution by year 10. The difference in the behavior to that of the EnKF could be related to the rapid adjustment of the biodegradation rates right after incorporating information about the background state-parameters cross-correlations.

---

## Author Comment (AC4) · 20 Jul 2016

[revised manuscript text omitted]

North sea

Hoek van Holland

Oostvoorne

Hellevoetsluis

Brielle

's-Gravenzande

Monster

Naaldwijk

De Lier

Wateringen

Maassluis

Maasland

Schipluiden

**Industrial areas**

Vlaardingen

Delft

**Polder areas**

Spijkenisse

**Polder areas**

Nieuw-Beijerland

Hoogvliet

Nieuwe Maas

**City of Rotterdam**

Antropogene upper layer

**POC 1**

**Holocene**

**POC 3**

**POC 3**

**POC 2**

Pleistocene aquifer

**Pleistocene deposits
with low permeable clay-rich upper layer**

[Figure]

[Figure]

[Figure]

[Figure]

[Figure]

**K-PCE cross-correlated with C-VC**

[Figure]

[Figure]

[Figure]

[Figure]

[Figure]

**Concentration Ensemble Statistics**

[Figure]

[Figure]

[Figure]

[Figure]

[Figure]

[Figure]

[Figure]

[Figure]

**Background Cross-Correlations: Changing Flow Conditions**

[Figure]

**Prediction Errors: EnKF-OI ($\alpha$)**

- $\alpha = 0.3$
- $\alpha = 0.4$
- $\alpha = 0.5$
- $\alpha = 0.6$
- $\alpha = 0.7$
- $\alpha = 0.8$

Using highly uncertain hydraulic parameters

Perfect flow conditions

Concentration ($\mu$g/l)

Years

**Performance of EnKF-OI (adaptive) and EnKF-ESOS**

Average MSE Difference

- Perfect Flow
- Uncertain Flow

K-PCE  K-TCE  K-DCE  K-VC

Biodegradation Rates

---

## Author Response (AR2)

*Dr. M. E. Gharamti*
*Mohn-Sverdrup Center for Operational Oceanography (NERSC)*
*C: (47) 947 222 09 E: mohamad.gharamti@nersc.no - Bergen - Norway*

September 26$^{th}$, 2016

Dear Dr. Hendricks Franssen,

Please find enclosed our revised manuscript "On the efficiency of the hybrid and the exact second-order sampling formulations of the EnKF: A reality-inspired 3D test case for estimating biodegradation rates of chlorinated hydrocarbons at the port of Rotterdam." The reviewer's comments were carefully considered in the revised manuscript. When we did not agree with his comments, we gave a thorough explanation. The reviewer's suggestion concerning the comparison between the serial and the original implementation of the adaptive hybrid scheme is very useful and we gratefully acknowledge it. Enclosed please also find a detailed response to the four points raised by the reviewer.

Essentially, we have compared the current serial implementation of the hybrid filter with the previously published one by Gharamti et al. (2014). We now further provide a discussion on the usefulness of the serial algorithm, compared to the original one, in section 4.1.2. The comparison is given in the new Figure 14.

We thank you for giving us a chance to resubmit a revised version of our manuscript and to reply to the reviewer's comments. We are also grateful for all your editorial efforts.

Please do not hesitate to contact me should you need anything else.

Sincerely yours,
M. E. Gharamti

*Dr. M. E. Gharamti*
*Mohn-Sverdrup Center for Operational Oceanography (NERSC)*
*C: (47) 947 222 09 E: mohamad.gharamti@nersc.no - Bergen - Norway*

**Reply to Reviewer 3**

We would like to thank the reviewer for his comments and suggestions. We followed the reviewer's recommendations and revised the manuscript accordingly. Our detailed replies to the reviewer's comments are given below.

1. I still have some concerns about the novelty of parts of the presented work. More than half of the results section deals with the comparison of EnKF and EnKF-OI and a sensitivity analysis on the weighting factors for EnKF-OI. This has already been presented in Gharamti et al. (2014) and I do not see why this discussion should be repeated in this manuscript (given that the conclusions and the overall model setup are very similar, see below). The authors should instead focus more on aspects that are built on top of the findings from Gharamti et al. (2014). For example, the authors mention that the sequential treatment of observations in the EnKF-OI scheme is a novelty in this work. However, if this is an innovative aspect of the paper why is this new method then not compared to the previous implementation? In the current state of the manuscript, the reader is left alone wondering if this change in the method really provides the supposed improvements.

The overall objective of the manuscript is to evaluate the contribution of the EnKF-OI and the $EnKF_{ESOS}$ to the analysis step of the standard EnKF scheme. We propose to implement the hybrid scheme serially and allow the adaptive algorithm to select different weighting factors for each individual observation. We run similar assimilation experiments, as in Gharamti et al. (2014), to test the validity of the new serial hybrid scheme. We agree with the reviewer that comparing the current algorithm to the previously published one may be useful. We therefore followed the reviewer's suggestion and compared the performance of the new scheme (serial hybrid) to the previous one (batch hybrid). We included a new figure (Fig. 14) and discussed it in a new paragraph at the end of section **4.1.2**
We found out that the serial algorithm performs slightly better than the original scheme. In terms of relative improvements, the batch and the serial hybrid schemes suggest around 52% and 57% more accurate state and parameter estimates than those of the standard EnKF. We also note that processing the observations serially leads to a smoother selection of the weights between the ensemble-based and the static background covariances. In the batch scheme, the optimization is relatively more erratic and exhibits stronger variations over time.
From an algorithmic point of view, optimizing the parameters $\alpha$ and $\beta$ in a serial sense is computationally more efficient and does not involve any matrix inversion, in contrast with the batch processing which requires the inversion of the matrix $\left( H_k P_k^f H_k^T + R_k \right)$ for every iteration of the optimization procedure

2. Additionally, a point that is stressed throughout the manuscript is that the authors intend to deal with a more realistic model setup. Real-world problem are generally characterized by uncertainty of subsurface parameters and boundary conditions, spatial heterogeneity, the presence of model structural errors and an increased level of complexity. However, most of the results presented in the paper are based on simplified model assumption, e.g., perfect knowledge on the flow field and hydraulic parameters and a lack of model

dynamics and heterogeneity. Due to these perfect model assumptions, the usage of a 3D model (instead of a 2D case) is also not likely to add much complexity to the overall setup. The new results including uncertainty in hydraulic parameters (presented now in section 4.3) are quite interesting and relevant for the practical application of these kinds of methods. So I would recommend the authors to extend these investigations and focus more on the practical issues of the method that arise under real-world conditions (as an ideal case was already discussed in their previous paper).

The objective of our manuscript differs from that of Gharamti et al. (2014). Apart from the hybrid's serial implementation, we investigated the impact/usefulness of accounting for sampling errors in both the forecast ensemble and the observations. The reviewer seems to miss this essential point and focuses his argument on the similarity to an earlier study. In the previous revision, we have followed the reviewer's recommendation and included a section where uncertainties are included in the groundwater flow component of the coupled system. Investigating other sources of uncertainty, such as those related to the boundary conditions, heterogeneity, structural errors, etc is interesting but does not fall within the scope of the current study. Building on the current findings about the data assimilation method, we will consider quantifying different sources of model uncertainties in a future study. This is now mentioned in the conclusion section. Thank you.

3. Currently, section 4.3 also lacks some quantitative information that underpin the comparison between EnKF and EnKF-OI. It is mentioned, that EnKF-OI outperforms EnKF for a moderate uncertainty of hydraulic parameters but no numbers are given to judge the different performance. Additionally, results for EnKF are missing in Figure 18.

We have now included the results from the EnKF using the perturbed flow scenarios. We report that under moderate flow-uncertainty, the EnKF-OI ($\alpha$= 0.7) and the EnKF-ESOS respectively yield 22% and 20% more accurate concentration estimates. This gain is obviously less important than the one obtained under perfect flow conditions. Figure 18 (now Fig. 19) has been updated to include the EnKF results, as suggested. Thank you.

4. Additionally, there is also a problem with the initialization of the assimilation experiments. In a real-world case, one would typically derive the initial concentration fields by simulating a warm up period with historical data until the assimilation of measurements starts at a certain time t0 (e.g., after a monitoring network was installed). However, what is done in this paper is that the initial concentration fields are derived by taking snapshots from a forward simulation for exactly the same time period that is used later in the assimilation experiments. The problem here is that the initial ensemble at time t0 already 'predicts' a considerable contamination along the whole (future) trajectory of the plume. On the contrary, the observations at time t0 state that the aquifer is completely clean initially. For a real-world situation this would be quite unrealistic because this huge discrepancy between observed and simulated concentration (clean versus totally contaminated) would heavily questions the credibility of the forward model, i.e. such a forward model would eventually be discarded due to the unrealistic bias. I would recommend

*Dr. M. E. Gharamti*
*Mohn-Sverdrup Center for Operational Oceanography (NERSC)*
*C: (47) 947 222 09 E: mohamad.gharamti@nersc.no - Bergen - Norway*

the authors to account for this problem and to adapt their model initialization accordingly.

We disagree with the reviewer's argument. In real-world situations, we often have little knowledge about the initial configuration of the model state variables and the distribution of the parameters. A biased initial ensemble is another complexity that, if present, the EnKF may struggle with. Furthermore, the free-run simulation (from which we collect snapshots to construct the initial ensemble) is subject to various sources of model uncertainties, as compared to the reference run. As mentioned in the manuscript, we perturb the initial conditions and the degradation parameters. Thus, the initial ensemble does not reflect directly the truth or the sampled observations.

Selecting snapshots from a perturbed run is not new. In fact, many operational and near-operational studies follow a similar strategy:

- Sakov, P., Counillon, F., Bertino, L., Lisæter, K.A., Oke, P.R. and Korablev, A., 2012. TOPAZ4: an ocean-sea ice data assimilation system for the North Atlantic and Arctic. Ocean Science, 8(4), p. 633.
- Hoteit, I., Hoar, T., Gopalakrishnan, G., Collins, N., Anderson, J., Cornuelle, B., Köhl, A. and Heimbach, P., 2013. A MITgcm/DART ensemble analysis and prediction system with application to the Gulf of Mexico. Dynamics of Atmospheres and Oceans, 63, pp.1-23.
- Counillon, F., Bethke, I., Keenlyside, N., Bentsen, M., Bertino, L. and Zheng, F., 2014. Seasonal-to-decadal predictions with the ensemble Kalman filter and the Norwegian Earth System Model: a twin experiment. Tellus A, 66.

The reviewer's suggestion for initializing is now acknowledged and mentioned in the revised manuscript (Section **3.4**). Thank you

---

## Author Response (AR3)

*Dr. M. E. Gharamti*
*Mohn-Sverdrup Center for Operational Oceanography (NERSC)*
*C: (47) 947 222 09 E: mohamad.gharamti@nersc.no - Bergen - Norway*

October 16[th], 2016

Dear Dr. Hendricks Franssen,

Please find enclosed our revised manuscript "On the efficiency of the hybrid and the exact second-order sampling formulations of the EnKF: A reality-inspired 3D test case for estimating biodegradation rates of chlorinated hydrocarbons at the port of Rotterdam." We have considered all your comments and suggestions. For the figures, we have merged a couple of them and removed Figure 17 from the older version. The total number is 15 now. The text has been revised accordingly.

We thank you for the time you spent editing this manuscript. We appreciate your patience and assistance. We are also grateful for all your editorial efforts.

Please do not hesitate to contact me should you need anything else.

Sincerely yours,
M. E. Gharamti